# SARS-CoV-2 spike spurs intestinal inflammation via VEGF production in enterocytes

Fa-Min Zeng[1,2,†], Ying-wen Li[1,†], Zhao-hua Deng[1,†], Jian-zhong He[2,†], Wei Li[2], Lijie Wang[1], Ting Lyu[2], Zhanyu Li[2], Chaoming Mei[1], Meiling Yang[1], Yingying Dong[1], Guan-Min Jiang[3], Xiaofeng Li[4] (iD), Xi Huang[5], Fei Xiao[1,5,*] (iD), Ye Liu[2,**] (iD), Hong Shan[1,6,***] (iD) & Huanhuan He[1,****] (iD)

## Abstract

Severe acute respiratory syndrome coronavirus-2 (SARS-CoV-2) can cause gastrointestinal (GI) symptoms that often correlate with the severity of COVID-19. Here, we explored the pathogenesis underlying the intestinal inflammation in COVID-19. Plasma VEGF level was particularly elevated in patients with GI symptoms and significantly correlated with intestinal edema and disease progression. Through an animal model mimicking intestinal inflammation upon stimulation with SARS-CoV-2 spike protein, we further revealed that VEGF was over-produced in the duodenum prior to its ascent in the circulation. Mechanistically, SARS-CoV-2 spike promoted VEGF production through activating the Ras-Raf-MEK-ERK signaling in enterocytes, but not in endothelium, and inducing permeability and inflammation. Blockage of the ERK/VEGF axis was able to rescue vascular permeability and alleviate intestinal inflammation *in vivo*. These findings provide a mechanistic explanation and therapeutic targets for the GI symptoms of COVID-19.

**Keywords** COVID-19; GI symptoms; intestinal inflammation; vascular permeability; VEGF
**Subject Categories** Digestive System; Microbiology, Virology & Host Pathogen Interaction; Vascular Biology & Angiogenesis

## Introduction

The coronavirus disease 2019 (COVID-19), caused by severe acute respiratory syndrome coronavirus-2 (SARS-CoV-2), has become a persistent health emergency since its outbreak in late 2019 (Rader *et al*, 2020). Since we and others provided initial evidence for gastrointestinal (GI) infection in COVID-19 patients (Jin *et al*, 2020; Lin *et al*, 2020; Xiao *et al*, 2020), the reports of GI manifestations caused by SARS-CoV-2 in clinical cases and nonhuman primate models have escalated (Jiao *et al*, 2021). Notably, GI symptoms are often correlated with disease severity and systemic inflammation (Jin *et al*, 2020; Wan *et al*, 2020).

It has been shown that specific insult of GI by SARS-CoV-2 can lead to systemic inflammation due to impaired GI barrier (Jiao *et al*, 2021). This was further exemplified by the fact that when the GI barrier was repaired by a zonulin antagonist, the systemic inflammation in Multisystem Inflammatory Syndrome in Children (MIS-C) upon SARS-CoV-2 infection or exposure was relieved (Gruber *et al*, 2020; Yonker *et al*, 2021). However, the pathogenesis of GI symptoms upon SARS-CoV-2 spike stimulation and how GI barrier impairment contributes to the systemic inflammation are still obscure, leaving treatment options nebulous.

Vascular barrier plays a key role in confining antigen spreading and inflammation. Increased inflammatory cytokines from intestine could overflow to bloodstream and cause lung inflammation (Mjösberg & Rao, 2018). Current studies proved that SARS-CoV-2 infection could induce vascular barrier dysfunction *in vivo* and *in vitro* (Colunga Biancatelli *et al*, 2021; Raghavan *et al*, 2021). However, as a critical component of GI barrier (Bouziat & Jabri, 2015), the role and the mechanism of intestinal vascular barrier in GI inflammation and disease progression are still unclear.

Progression of COVID-19 often involves excessive proinflammatory cytokines and mediators (Choreño-Parra *et al*, 2021). VEGF, a key factor involved in vascular permeability and inflammation (Lee *et al*, 2004), was found skyrocketed in the blood of

1 Guangdong Provincial Key Laboratory of Biomedical Imaging and Guangdong Provincial Engineering Research Center of Molecular Imaging, The Fifth Affiliated Hospital, Sun Yat-sen University, Zhuhai, China
2 Department of Pathology, The Fifth Affiliated Hospital, Sun Yat-sen University, Zhuhai, China
3 Department of Clinical Laboratory, The Fifth Affiliated Hospital, Sun Yat-sen University, Zhuhai, China
4 Department of Gastroenterology, The Fifth Affiliated Hospital, Sun Yat-sen University, Zhuhai, China
5 Department of Infectious Diseases, The Fifth Affiliated Hospital, Sun Yat-sen University, Zhuhai, China
6 Department of Interventional Medicine, The Fifth Affiliated Hospital, Sun Yat-sen University, Zhuhai, China
*Corresponding author. Tel: (+86)756-2528024; E-mail: xiaof35@mail.sysu.edu.cn
**Corresponding author. Tel: (+86)756-2528106; E-mail: ly77219@163.com
***Corresponding author. Tel: (+86) 756-2528555; E-mail: shanhong@mail.sysu.edu.cn
****Corresponding author. Tel: (+86)756-2526143; E-mail: hehh23@mail.sysu.edu.cn
†These authors contributed equally to this work

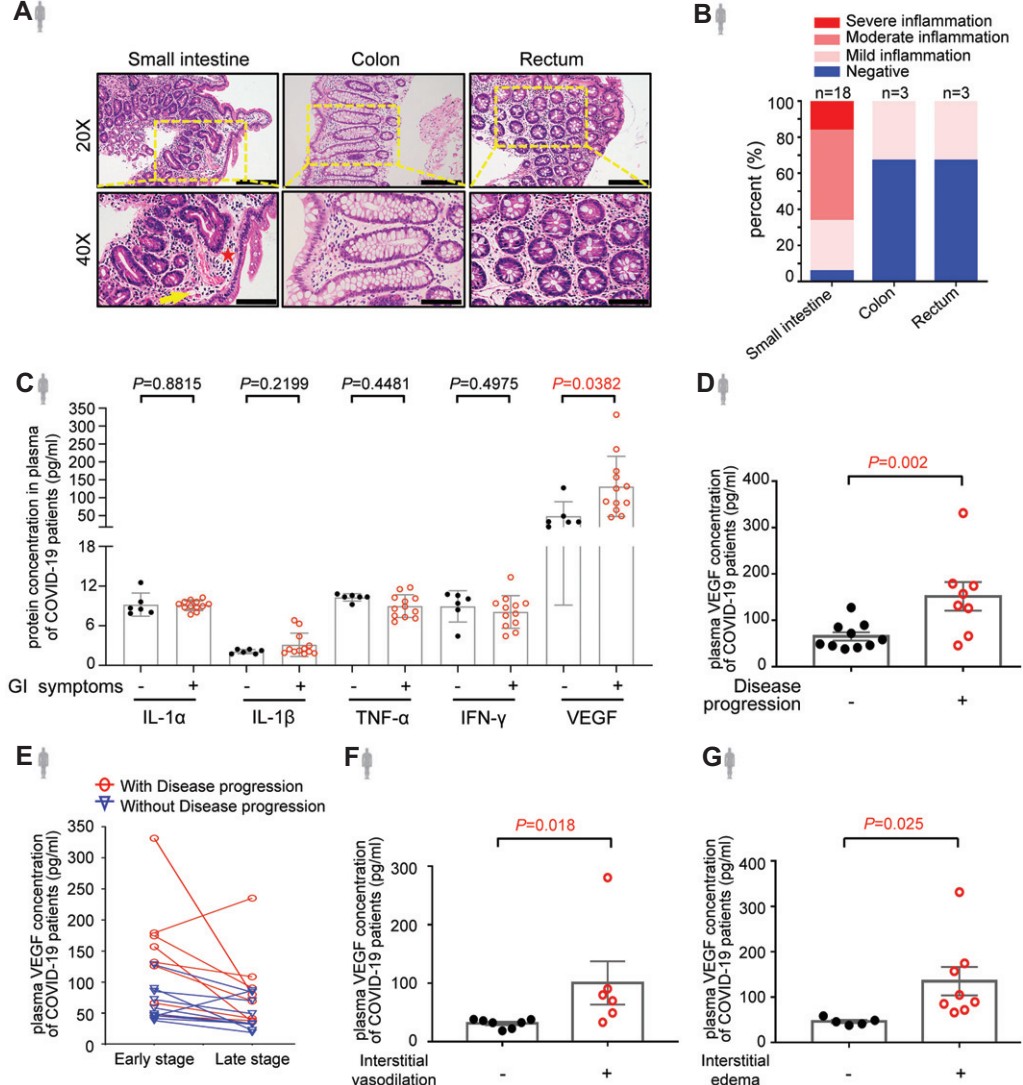

**Figure 1. Plasma VEGF level correlates with GI symptoms and disease progression of COVID-19.**

A H&E staining of sections of intestinal tissues obtained from endoscopic biopsy of COVID-19 patients who presented with gastrointestinal (GI) symptoms. The inflammatory infiltrates were indicated by a yellow arrow. The edema area was indicated by a red star. Scale bars, 100 μm.

B The bar chart shows the degrees of intestinal inflammation in patients with COVID-19. Number of samples for each group as indicated.

C Cytokine levels in the plasma of COVID-19 patients with ($n = 12$) or without ($n = 6$) GI symptoms by cytokine assay.

D Levels of plasma VEGF in COVID-19 patients with ($n = 8$) or without ($n = 10$) disease progression by cytokine assay.

E Temporal course of plasma VEGF at the early and late stage of COVID-19 infection by cytokine assay. Data shown are the levels of plasma VEGF in patients with ($n = 8$) and without ($n = 10$) disease progression at the early stage (one to three days after laboratory-confirmed for COVID-19) and late stage (more than three days after laboratory-confirmed for COVID-19).

F Levels of plasma VEGF in COVID-19 patients with ($n = 6$) or without ($n = 7$) interstitial vasodilation by cytokine assay.

G Levels of plasma VEGF in COVID-19 patients with ($n = 8$) or without ($n = 5$) interstitial edema by cytokine assay.

Data information: All data are shown as mean ± SD. $P$ values are determined by Student's $t$-test.
Source data are available online for this figure.

COVID-19 patients and related to disease severity (Polidoro et al, 2020; Syed et al, 2021). Based on the crucial role of VEGF in pulmonary edema and inflammation upon SARS-CoV-2 infection in the lung, a clinical trial has made its successful debut in treating severe lung injury of COVID-19 patients by targeting VEGF with Bevacizumab (Pang et al, 2021). However, the role of VEGF in SARS-CoV-2-mediated GI inflammation remains enigmatic.

The spike protein of SARS-CoV-2 mediates the binding of virus to host cell receptors or co-receptors including angiotensin-converting enzyme 2 (ACE2), transmembrane protease serine 2 (TMPRSS2), and neuropilin-1 (NRP-1) (Hoffmann et al, 2020; Mayi et al, 2021). Besides, the spike protein can alter cellular functions by activating various signaling pathways (Lei et al, 2021; Moutal et al, 2021). Although it has been reported that spike protein could induce lung

**Table 1. The correlation between interstitial edema/ vasodilation and GI manifestations in 18 patients with SARS-CoV-2 infection.**

| | Interstitial edema | | Interstitial vasodilation | |
|---|---|---|---|---|
| | *R* | *P* | *R* | *P* |
| Disease classification (Non-severe, Severe) | 0.544 | 0.001 | 0.172 | 0.322 |
| Symptoms | | | | |
| Diarrhea | 0.151 | 0.386 | −0.180 | 0.302 |
| Anorexia | −0.187 | 0.282 | −0.124 | 0.478 |
| Nausea | −0.320 | 0.061 | 0.180 | 0.300 |
| Vomit | −0.258 | 0.134 | 0.241 | 0.164 |
| Acid reflux | 0.375 | 0.027 | 0.049 | 0.779 |
| Epigastric discomfort | 0.180 | 0.302 | 0.112 | 0.521 |
| Hepatic function impairment | | | | |
| Total bilirubin (µmol/l; normal range 3.0–24.0) | 0.374 | 0.027 | 0.030 | 0.866 |
| ALT (U/l; normal range 7–40 in female, 9–50 in male) | 0.475 | 0.004 | 0.215 | 0.214 |
| AST (U/l; normal range 13–35 in female, 15–40 in male) | 0.487 | 0.003 | 0.323 | 0.059 |

ALT, alanine aminotransferase; AST, aspartate aminotransferase (value on initial presentation).

$P < 0.05$ was considered statistically significant (in red).

**Table 2. The correlation between plasma VEGF and clinicopathological features in patients with COVID-19.**

| | VEGF (Test cohort) | | VEGF (Validation cohort) | |
|---|---|---|---|---|
| | *R* | *P* | *R* | *P* |
| Disease progression | 0.490 | 0.002 | 0.756 | <0.001 |
| Interstitial edema | 0.460 | 0.018 | | |
| Interstitial vasodilation | 0.439 | 0.025 | | |
| Hepatic function impairment | | | | |
| Total bilirubin (µmol/l; normal range 3.0–24.0) | 0.677 | <0.001 | 0.507 | 0.027 |
| ALT (U/l; normal range 7–40 in female, 9–50 in male) | 0.51 | 0.011 | 0.728 | <0.001 |
| AST (U/l; normal rage 13–35 in female, 15–40 in male) | 0.566 | 0.004 | 0.492 | 0.032 |
| Serologic markers of disease severity | | | | |
| CRP (µg/l; normal range 0.068–8.2) | 0.280 | 0.261 | 0.582 | 0.004 |
| D-dimers (µg/l; normal range 0–243) | −0.104 | 0.687 | 0.404 | 0.086 |
| Procalcitonin | 0.236 | 0.346 | 0.579 | 0.009 |
| Nucleic acid test results of fecal samples (Positive or Negative) | −0.086 | 0.733 | −0.005 | 0.983 |

$P < 0.05$ was considered statistically significant (in red).

injury and barrier dysfunction in pulmonary endothelial cells (Colunga Biancatelli *et al*, 2021), whether and how the viral spike protein alone contributes to the GI inflammation and gut vascular barrier stays unknown.

Here, we investigated the underlying mechanisms involved in SARS-CoV-2 spike-induced VEGF production and inflammation in the intestine, which may provide therapeutic targets for the remediation of GI symptoms in patients with COVID-19.

# Results

### Serum VEGF level is elevated in COVID-19 patients with GI symptoms and correlated with intestinal edema and disease progression

To systemically characterize intestinal infection and inflammation in COVID-19 patients, we evaluated different segments of intestinal tissues from 18 patients who were diagnosed with COVID-19 and presented with GI symptoms (Table EV1). Robust expression of virus receptor ACE2 and virus antigen spike protein was detected in small intestine (Fig EV1A and B), indicating SARS-CoV-2 infection in GI tract, which agreed with previous studies (Du *et al*, 2020; Zhang *et al*, 2020). Intriguingly, small intestine displayed the most severe inflammation characterized by inflammatory cell infiltration and signs of vascular damage such as interstitial edema, hemorrhage and vasodilation (Fig 1A and B). Notably, interstitial edema was significantly correlated with disease type, acid reflux, total bilirubin, alanine aminotransferase (ALT), and aspartate aminotransferase (AST) (Table 1 and Table EV1), implying vascular permeability change as a crucial factor in disease progression. Since it has been reported that myriad cytokines were raised in the blood of COVID-19 patients (Mangalmurti & Hunter, 2020), we examined the ones that could potentially affect permeability, including IL-1α, IL-1β, TNF-α, IFN-γ, and VEGF, in the cases with or without GI symptoms. Intriguingly, only VEGF level showed significant elevation in the plasma of patients with GI symptoms compared to those without (Fig 1C). Subsequently, we expanded the serologic analysis of VEGF concentrations in another independent cohort and confirmed the correlation between VEGF level and GI symptoms (Fig EV1C). Not only was VEGF correlated with disease progression (Figs 1D and EV1D, and Table 2), but the VEGF level was even higher at the early stage of the progressed patients (Figs 1E and EV1E), suggesting that VEGF may lead the cytokine cascade and contribute to the inflammation in the early phase of viral infection. To be noticed, the VEGF level was also significantly associated with gut interstitial edema and vasodilation (Fig 1F and G), restating the role of VEGF in intestinal inflammation.

### VEGF is induced in intestinal tissue upon abdominal stimulation of SARS-CoV-2 spike

Next, we asked where plasma VEGF was from with the speculation that it could be induced in the local inflammatory tissue. Indeed, we detected a higher VEGF-A level in the intestinal tissues from COVID-19 patients compared to those from healthy individuals (Fig 2A). To

be noted, only VEGF-A was significantly raised, leaving VEGF-B and VEGF-C and VEGF receptors unaltered (Fig EV1F and G). In order to validate whether SARS-CoV-2 was responsible for the VEGF production in intestinal tissue, we generated an animal model to specifically mimic the intestinal inflammation by intraperitoneally injecting the recombinant spike-Fc containing the receptor binding domain (RBD) to C57BL/6J mice (Fig EV2A). Co-localization of murine ACE2 with spike was confirmed by immunofluorescence staining (Fig EV2B–D), which was consistent with previous results (Kuba *et al*, 2005; Raghavan *et al*, 2021; Shin *et al*, 2021). Since

intestinal mucosal barrier renders strong protection from the infectious agents (Sharma & Riva, 2020), treating mice with spike RBD alone for a short time only gave rise to mild inflammation (Fig EV2E). Hence, we treated the intestinal epithelium by acid enema for a very short time (2 min) prior to viral challenge adapted from a previous lung infection model (Kuba *et al*, 2005) (Fig 2B). Similar to the observations in COVID-19 patients, treatment by SARS-CoV-2 spike RBD induced intestinal inflammation with the most severe pathological alterations in the duodenum (Fig 2C and D; Table 3). Overproduced inflammatory factors, such as IL-1α,

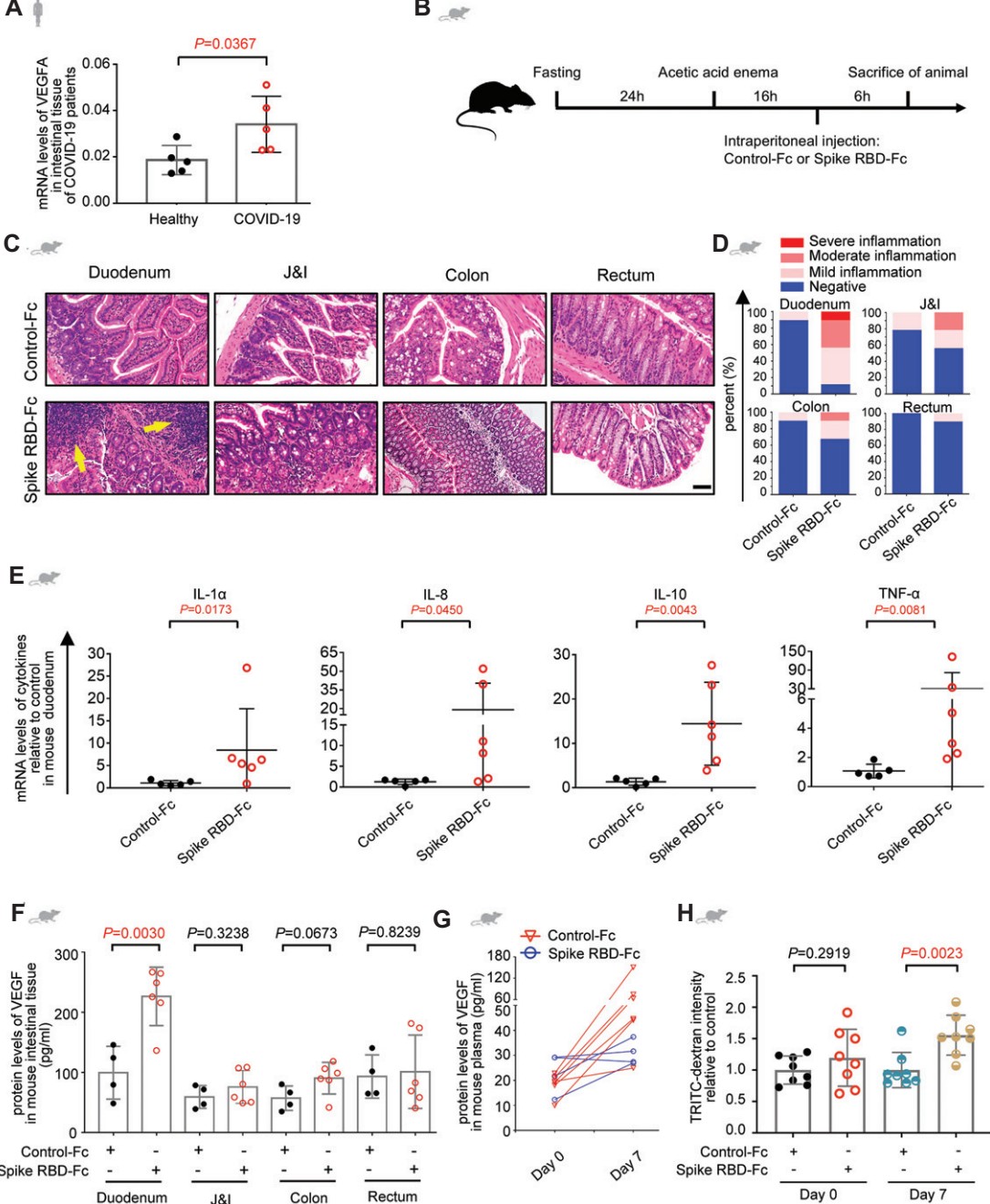

**Figure 2.**

**Figure 2. VEGF is elevated in small intestine upon abdominal stimulation of SARS-CoV-2 spike.**

A Levels of VEGF-A mRNA in the intestinal tissues from COVID-19 patients (n = 5) or healthy controls (n = 5) by RNA-seq.

B Illustration depicts generation of the animal model with SARS-CoV-2 spike RBD-induced intestinal inflammation. Mice were fasted for 24 h before acetic acid enema. 16 h after acid challenge, mice were given Control-Fc or Spike RBD-Fc intraperitoneally. Six hours later, TRITC-dextran or Evans Blue dye was injected intravenously for evaluation of permeability.

C, D Representative H&E images (C) and the quantitative analysis of inflammation (D) of the intestinal tissues from animals treated by the protocol shown in (B). The inflammatory infiltrates were indicated by a yellow arrow. Scale bar, 100 μm. Control-Fc, n = 5; Spike RBD-Fc, n = 6. n, biologically independent samples (mice).

E qRT–PCR analysis of levels of inflammatory factors in duodenum tissues of mice treated with Control-Fc or Spike RBD-Fc. Control-Fc, n = 5; Spike RBD-Fc, n = 6. n, biologically independent samples (mice).

F ELISA analysis of VEGF concentration in the intestinal tissues of mice treated with Control-Fc or Spike RBD-Fc. Control-Fc, n = 4; Spike RBD-Fc, n = 6. n, biologically independent samples (mice).

G ELISA analysis of VEGF concentration in the plasma of mice after six hours after treatment with Control-Fc or Spike RBD-Fc, or seven days after treated with Control-Fc or Spike RBD-Fc daily. Control-Fc, n = 4; Spike RBD-Fc, n = 6. n, biologically independent samples (mice).

H Evaluation of murine peritoneal permeability six hours after treatment with Control-Fc or Spike RBD-Fc, or seven days after treated with Control-Fc or Spike RBD-Fc daily by TRITC-dextran dye extravasation assay. For each group, n = 8. n, biologically independent samples (mice).

Data information: All data are shown as mean ± SD. For (A), (F), and (H), *P* values are determined by Student's *t*-test; for (E), *P* values are determined by Mann–Whitney test. Source data are available online for this figure.

**Table 3. Inflammatory changes and interstitial edema in intestinal tissue of mice treated with SARS-CoV-2 Spike RBD protein.**

| | | C57BL/6J Mouse (Test cohort) | | hACE2-B6J Mouse (Validation cohort) | |
| --- | --- | --- | --- | --- | --- |
| | | Control-Fc (n = 5) | Spike-Fc (n = 6) | Control-Fc (n = 6) | Spike-Fc (n = 6) |
| **Inflammatory changes** | | | | | |
| Duodenum | − | 4 (80%) | 0 (0%) | 5 (83.3%) | 0 (0%) |
| | + | 1 (20%) | 3 (50%) | 1 (16.7%) | 2 (33.3%) |
| | ++ | 0 (0%) | 2 (33.3%) | 0 (0%) | 4 (66.7%) |
| | +++ | 0 (0%) | 1 (16.7%) | 0 (0%) | 0 (0%) |
| J&I | − | 5 (100%) | 1 (16.7%) | 5 (83.3%) | 1 (16.7%) |
| | + | 0 (0%) | 5 (83.3%) | 1 (16.7%) | 3 (50%) |
| | ++ | 0 (0%) | 0 (0%) | 0 (0%) | 2 (33.3%) |
| | +++ | 0 (0%) | 0 (0%) | 0 (0%) | 0 (0%) |
| Colon | − | 4 (80%) | 3 (50%) | 5 (83.3%) | 4 (66.7%) |
| | + | 1 (20%) | 2 (33.3%) | 1 (16.7%) | 2 (33.3%) |
| | ++ | 0 (0%) | 1 (16.7%) | 0 (0%) | 0 (0%) |
| | +++ | 0 (0%) | 0 (0%) | 0 (0%) | 0 (0%) |
| Rectum | − | 4 (80%) | 3 (50%) | 6 (100%) | 5 (83.3%) |
| | + | 1 (20%) | 2 (33.3%) | 0 (0%) | 1 (16.7%) |
| | ++ | 0 (0%) | 1 (16.7%) | 0 (0%) | 0 (0%) |
| | +++ | 0 (0%) | 0 (0%) | 0 (0%) | 0 (0%) |
| **Interstitial edema** | Duodenum | 0 (0%) | 6 (100%) | 0 (0%) | 6 (100%) |
| | J&I | 0 (0%) | 4 (66.7%) | 0 (0%) | 3 (50%) |
| | Colon | 1 (20%) | 4 (66.7%) | 2 (33.3%) | 1 (16.7%) |
| | Rectum | 3 (60%) | 3 (50%) | 2 (33.3%) | 2 (33.3%) |

−, Negative; +, Mild inflammation; ++, Moderate inflammation; +++, Severe inflammation.

IL-8, IL-10, and TNF-α, specifically in the duodenum further confirmed the tissue tropism to the sites of inflammation (Figs 2E and EV2F). Since it is still controversial whether spike RBD can bind to murine ACE2 (Wu *et al*, 2020; Niu *et al*, 2021), spike-induced hACE2-B6J mice were adopted as confirmatory animal models and similar phenotype was observed (Fig EV3A and B). Intriguingly, consistent with our speculation but more specific, levels of murine VEGF in the duodenum of spike-stimulated C57BL/6J or hACE2-B6J mice were significantly raised (Figs 2F, and EV3C and D). More importantly, plasma VEGF was not elevated initially, but markedly increased after seven days of continual treatment of spike RBD (Figs 2G and EV3E). Moreover, the abdominal vascular hyperpermeability echoed the findings from patient tissues (Figs 2H and EV3F). The above results indicate that the initial VEGF could originate from local production in the small intestine, which may lead to systemic VEGF promotion over long-term inflammation.

## SARS-CoV-2 spike spurs VEGF production in enterocytes via the Ras-Raf-MEK-ERK pathway

In order to pinpoint the cellular source of VEGF in the intestine, human umbilical vein endothelial cells (HUVECs) were incubated with spike RBD, but yielded no obvious change of VEGF expression (Fig 3A). However, when spike RBD was added to the enterocyte Caco-2, the VEGF production was strikingly boosted (Fig 3B). Furthermore, the results of qRT–PCR for VEGF family proteins detected a significant elevation of VEGF-A, VEGF-B and VEGF-D with VEGF-C unchanged (Fig EV4A). The immunofluorescence analysis of tissues from patients and spike-stimulated hACE2-B6J mice also revealed co-localization of VEGF and enterocytes (Figs 3C and EV4B), which confirmed the VEGF production in enterocytes.

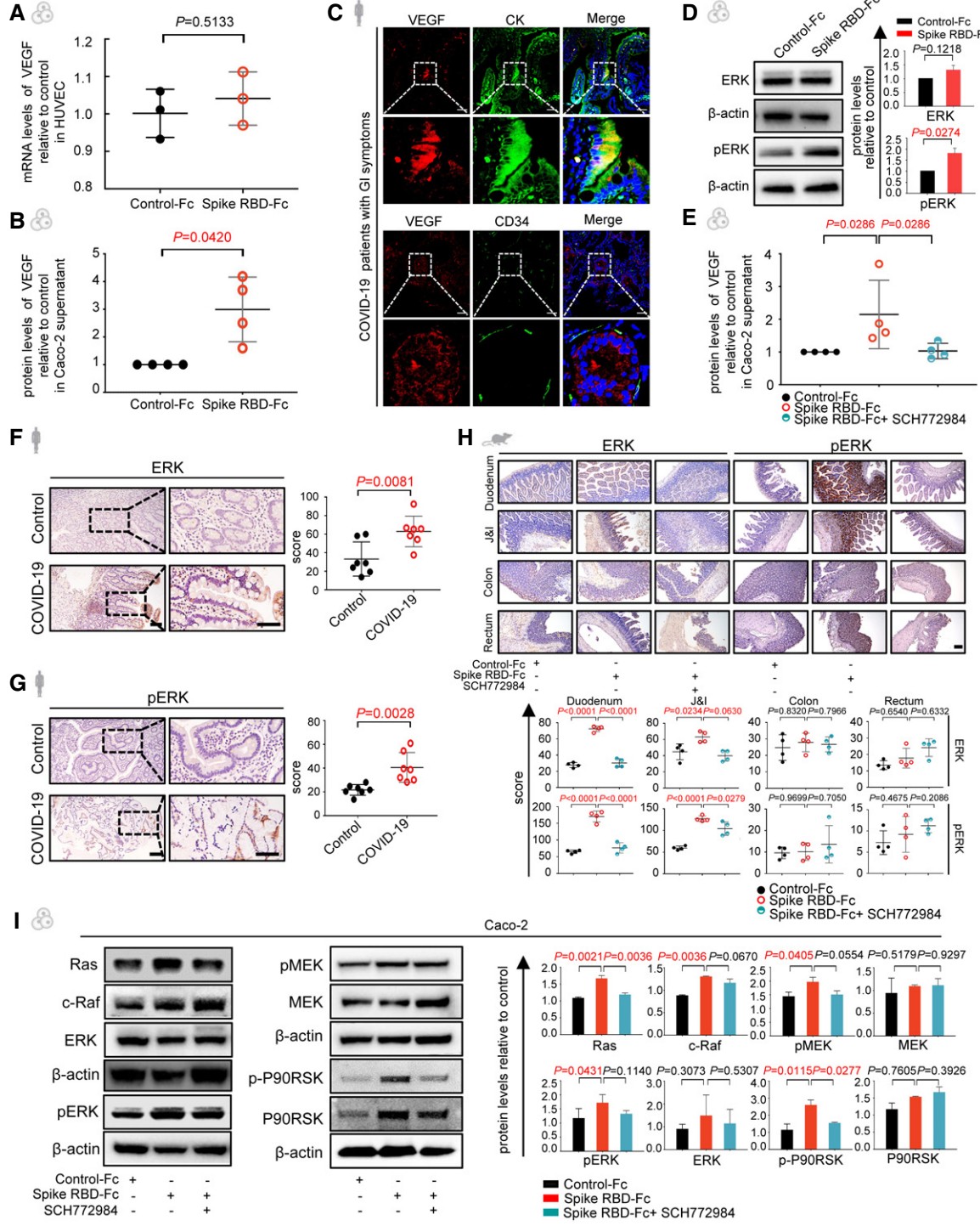

**Figure 3.**

**Figure 3.  SARS-CoV-2 spike RBD promotes VEGF production via the Ras-Raf-MEK-ERK pathway.**

A    qRT–PCR analysis of levels of VEGF mRNA in HUVEC treated with Control-Fc or Spike RBD-Fc. For each group, *n* = 3. Data are from three independent biological experiments.

B    ELISA analysis of VEGF concentration in the supernatant from Caco-2 cells treated with Control-Fc or Spike-Fc. For each group, *n* = 4. Data are from four independent biological experiments.

C    Immunofluorescence analysis of VEGF, cytokeratin (CK) and CD34 protein in the intestinal tissues of COVID-19 patients. CK and CD34 staining in green, VEGF staining in red, and nuclear staining in blue. Scale bars, 50 μm.

D    Levels of ERK and pERK in Caco-2 cells treated with Control-Fc or Spike RBD-Fc by Western blot. Protein expression was normalized to β-actin. Data are from three independent biological experiments.

E    ELISA analysis of VEGF concentration in the supernatant from Caco-2 cells treated with Control-Fc, Spike RBD-Fc, or Spike RBD-Fc combined with SCH772984. For each group, *n* = 4. Data are from four independent biological experiments.

F, G  Immunohistochemical staining shows ERK (E) and pERK (F) expression in the duodenum of COVID-19 patients and healthy controls. Scale bars, 100 μm. For each group, *n* = 7. *n*, biologically independent samples (human specimen).

H    Immunohistochemical staining shows ERK and pERK expression in the intestinal tissues of mice treated with Control-Fc, Spike RBD-Fc or Spike RBD-Fc combined with SCH772984. Scale bar, 100 μm. For each group, *n* = 4. *n*, biologically independent samples (human specimen).

I    The levels of Ras, c-Raf, MEK, pMEK, ERK, pERK, P90RSK, and p-P90RSK in Caco-2 cells treated with Control-Fc, Spike RBD-Fc, or Spike RBD-Fc combined with SCH772984 by Western blot. Protein expression was normalized to β-actin. Data are from three independent biological experiments.

Data information: All data are shown as mean ± SD. For (A), (B), (D), *P* values are determined by Paired Student's *t*-test; (F) and (G), *P* values are determined by Student's *t*-test; for (E), *P* values are determined by Mann–Whitney test; for (H) and (I), *P* values are determined by one-way ANOVA.

Source data are available online for this figure.

Hence, we investigated the underlying mechanism involved in the spike-mediated VEGF production in enterocytes.

It has been reported in several scenarios that activating ERK signaling pathway can lead to the over-expression of VEGF in human colonocytes (Huang *et al*, 2019). Therefore, we examined this pathway in Caco-2 cells exposed to spike RBD. Western blot detected a reliable activation of phosphorylated ERK (Fig 3D), while the expression of other important pathways such as phosphorylation of AKT was not affected (Fig EV4C). Furthermore, VEGF production could be blocked by administering the ERK inhibitor, SCH772984 (Fig 3E). In line with the previous result, the ERK pathway in spike RBD-treated HUVECs remained unchanged (Fig EV4D). Importantly, the alterations of ERK and pERK were also validated in COVID-19 patient specimens (Fig 3F and G) and animal tissues (Fig 3H).

Furthermore, we tested the possible upstream molecules of ERK in spike RBD-treated Caco-2 and identified the activation of the Ras-Raf-MEK-ERK pathway based on previous research (Chen *et al*, 2010) (Figs 3I and EV4E). These results were validated in a normal intestinal epithelial cell line, NCM460 (Fig EV4F). We next explored whether this pathway depended on ACE2. A dip of ACE2 protein level was observed in spike RBD-treated Caco-2 but not in HUVECs (Fig EV4G), which was consistent with the literature (Kuba *et al*, 2005) and the alternations of the downstream pathway. To further prove that the activation of the Ras-Raf-MEK-ERK cascade by spike RBD was ACE2-dependent, we knocked down ACE2 and found the phosphorylation of ERK induced by spike RBD did not further increase (Fig EV4H and I), indicating that ACE2 is essential to activate the spike RBD-mediated Ras-Raf-MEK-ERK signaling in the enterocytes. In summary, these findings define the critical role of ERK signaling pathway in spike RBD-induced enteric VEGF production, which may be mediated by ACE2.

## SARS-CoV-2 spike disrupts vascular barrier through attenuating VE-cadherin of the endothelium

So far, we have demonstrated that the SARS-CoV-2 spike can activate ERK/VEGF signaling in enterocytes, which induces inflammation and vascular hyperpermeability in the intestine *in vivo*. To assure that spike RBD-stimulated enterocytes could induce vascular permeability, HUVECs were incubated with the conditional medium of spike RBD-treated intestinal epithelial cells. Permeability assay detected a notably leaky endothelium when exposed to the supernatant of spike RBD-stimulated Caco-2 compared to the control (Fig 4A). To ascertain the molecules involved in such regulation, we examined the tight junctions and adherens junctions composing the vascular barrier (Spadoni *et al*, 2015). Western blot revealed no obvious changes in tight junctions such as ZO-1 (Fig 4B). However, the expression of VE-cadherin (VE-cad), a key adherens junction protein, was decreased, accompanied by an increased phosphorylation (pVE-cad) on the Tyr731 site, but not on the Tyr658 site (Fig 4B). We confirmed such changes in the intestinal tissues from COVID-19 patients (Fig 4C and D).

In order to evaluate the efficacy of blocking the ERK/VEGF axis in the rescue of vascular barrier, we introduced SCH77298, an ERK inhibitor, and Bevacizumab, an anti-VEGF antibody, to the enterocyte-EC co-culture separately. Either SCH772984 or Bevacizumab alone could significantly inhibit the internalization and phosphorylation of VE-cad (Figs 4E and EV5A) and rescue the spike RBD-induced vascular leakiness (Fig 4F). To confirm these findings *in vivo*, either SCH77298 or Bevacizumab was given to spike RBD-treated animals and VE-cad level was restored by boosting total VE-cad and dampening pVE-cad (Fig 4G). These data suggest that hindering the ERK/VEGF pathway could restore the endothelial barrier disrupted by spike RBD.

## Blockage of ERK/VEGF alleviates spike-induced hyperpermeability and inflammation

We then investigated the potentiality of targeting the ERK/VEGF pathway to improve SARS-CoV-2 spike RBD-induced hyperpermeability and inflammation in the pre-clinical setting. Consistent with the above findings, VEGF overproduction in the spike RBD-treated animals was abolished by either SCH772984 or Bevacizumab (Fig 5A), accompanied with downmodulation of ERK/pERK in the duodenum tissue (Fig EV5B). We next determined the leakiness of the vascular barrier

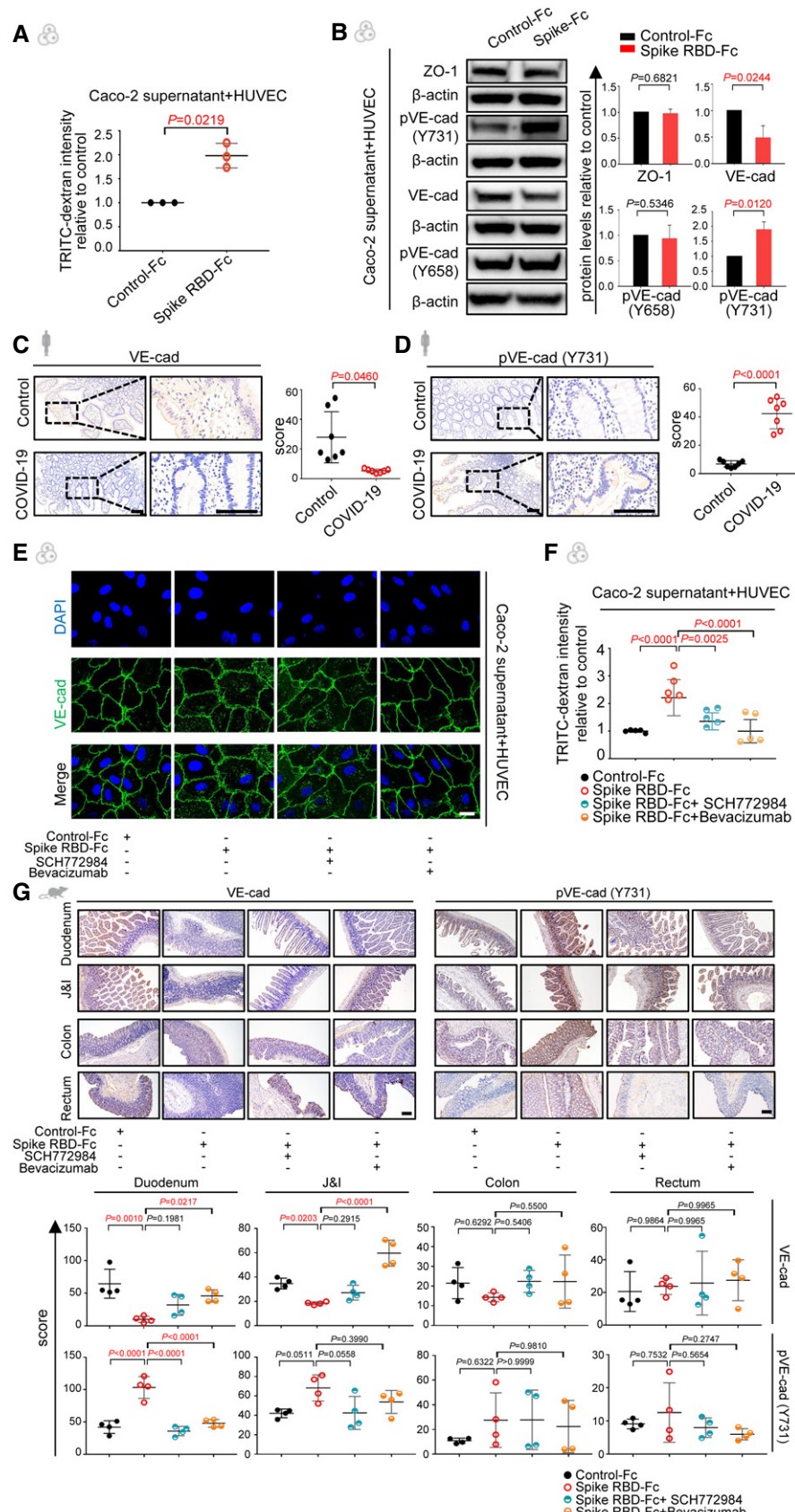

**Figure 4.**

**Figure 4. SARS-CoV-2 spike protein disrupts the vascular barrier via inhibiting endothelial VE-cadherin.**

A   The normalized fluorescence intensity of TRITC-Dextran dye from co-culture systems in which Caco-2 cells were incubated with Control-Fc or Spike RBD-Fc after 24 h. For each group, $n = 3$. Data are from three independent biological experiments.
B   The expression of junctional proteins ZO-1, VE-cadherin (VE-cad), phosphorylated VE-cad (pVE-cad; 658), and pVE-cad (731) in HUVECs co-cultured with the supernatant of Caco-2 cells that were treated with either Control-Fc or Spike RBD-Fc detected by Western blot. Protein expression was normalized to β-actin. Data are from three independent biological experiments.
C, D   Immunohistochemical staining shows levels of VE-cad (C) and pVE-cad (731) (D) in the duodenum tissue from COVID-19 patients and healthy controls. Scale bars, 100 μm. For each group, $n = 7$. $n$, biologically independent samples (human specimen).
E   Immunofluorescence analysis of VE-cad expression and subcellular localization in Caco-2 cells that were treated with Control-Fc, Spike RBD-Fc, or Spike RBD-Fc combined with SCH772984 or Bevacizumab by confocal microscopy. VE-cad staining in green and nuclear staining in blue. Scale bar, 20 μm.
F   Fluorescence intensity of TRITC-Dextran dye from co-culture systems in which Caco-2 cells were incubated with Control-Fc, Spike RBD-Fc, or Spike RBD-Fc combined with SCH772984 or Bevacizumab. Data are from five technical replicates with similar results from three biological replicates.
G   Immunohistochemical staining shows levels of VE-cad and pVE-cad (731) in the intestinal tissues of mice treated with Control-Fc, Spike RBD-Fc, or Spike RBD-Fc combined with SCH772984 or Bevacizumab. Scale bar, 100 μm. For each group, $n = 4$. $n$, biologically independent samples (mice).

Data information: All data are shown as mean ± SD. For (A) and (B), $P$ values are determined by paired Student's $t$-test; for (C) and (D), $P$ values are determined by Student's $t$-test; for (F) and (G), $P$ values are determined by one-way ANOVA.
Source data are available online for this figure.

upon either SCH772984 or Bevacizumab treatment by Evans Blue assay. The amount of Evans Blue dye leaked into the duodenum tissue from the SCH772984 or Bevacizumab treated group was significantly reduced, almost identical to the Control-Fc treated group (Fig 5B). Pathologically, the duodenum was less inflamed upon SCH772984 or Bevacizumab treatment (Fig 5C and D). We also repeated the experiment using the antibody against murine VEGF (Bock *et al*, 2009) and retrieved similar results (Fig EV5C–F). Together, the above results demonstrate that inhibition of the ERK/VEGF axis could rescue the compromised vascular barrier and alleviate the intestinal inflammation induced by spike RBD, highlighting a novel strategy to treat GI symptoms of COVID-19.

# Discussion

Here, we uncovered the association of VEGF with GI inflammation and disease progression in COVID-19 patients. Our data also identified a possible route and mechanism of SARS-CoV-2 spike-induced VEGF production and vascular permeability that might further develop and contribute to systemic inflammation. More specifically, SARS-CoV-2 spike boosts VEGF level by activating the Ras-Raf-MEK-ERK pathway in enterocytes, which results in VE-cad-mediated vascular hyperpermeability and ultimately leads to local and systemic inflammation. Blocking either ERK or VEGF has been able to relieve the intestinal inflammation stimulated by spike *in vivo* (Fig 5E). This study highlights the mechanism underlying the spike-activated ERK/VEGF pathway and intestinal inflammation, providing potential therapeutic targets for the mitigation of GI symptoms to improve the overall condition of COVID-19 patients.

Gastrointestinal symptoms are common manifestations of COVID-19 and have been identified as the key factors of severity and morbidity. Our investigation on VEGF advances the understanding of its association with GI symptoms and disease progression in two aspects. Firstly, as a pre-inflammatory factor, VEGF participates in intestinal inflammation response. To be noticed, VEGF level in blood of the animal model was not increased at the initial stage of intestinal inflammation, but was significantly boosted upon continuous treatment of spike RBD, suggesting a local to systemic spread that could further trigger more cytokines

release and systemic damage. Secondly, as a vascular permeability factor, VEGF induces local vascular leakage, which may further result in extravasated inflammatory cytokines. Consistent with our hypothesis, a recent study reported that the repair of gut mucosa barrier successfully restricted the cytokine storm and multisystem inflammation of MIS-C patients (Yonker *et al*, 2021). Our study confirms the protective role of gut vascular barrier in COVID-19. However, there may exist other SARS-CoV-2 targeted organs or cells that can produce VEGF, hence contributing to the systemic injury (Xu *et al*, 2020). Additionally, our model only investigated the role of SARS-CoV-2 spike and may not reflect the actual effects of SARS-CoV-2 virus on intestinal endothelium, which might account for the fact that injury of intestinal vascular barrier was indeed observed in COVID-19 patients with unclear mechanisms (Varga *et al*, 2020).

Enterocytes have been one of the most studied targets of SARS-CoV-2 mainly due to their abundant expression of ACE2. It has been reported that deficiency of ACE2 in digestive system links to dysregulation of nutrient transportation or renin-angiotensin system, which consequently participates in intestinal inflammatory response (Hashimoto *et al*, 2012; Penninger *et al*, 2021). Here, we provide an alternative route by showing that the ERK/VEGF axis in enterocytes plays a pivotal role in SARS-CoV-2-induced intestinal inflammation. However, we have no adequate evidence to support the mediating role of ACE2 in the SARS-CoV-2 spike-induced effects since the affinity of murine ACE2 and SARS-CoV-2 spike is still debatable (Li *et al*, 2004; Kuba *et al*, 2005; Zhao *et al*, 2020; Gu *et al*, 2021; Nuovo *et al*, 2021; Raghavan *et al*, 2021; Shin *et al*, 2021; Wang *et al*, 2021). We consider that whether the effect of spike RBD is through or fully through ACE2 is beyond the scope of this work. The fact that mice with either murine ACE2 or human ACE2 had similar phenotype might hint that an alternative receptor or, more likely, a co-factor of ACE2 is involved (preprint: Gu *et al*, 2020; Nguyen *et al*, 2022). Detailed mechanisms underlying this effect would worth further investigations. Taken together, our study defines the molecular mechanism of VEGF overproduction in SARS-CoV-2 spike-stimulated intestine and identifies ERK/VEGF as potential biomarkers and therapeutic targets of the intestinal inflammation and disease progression of COVID-19.

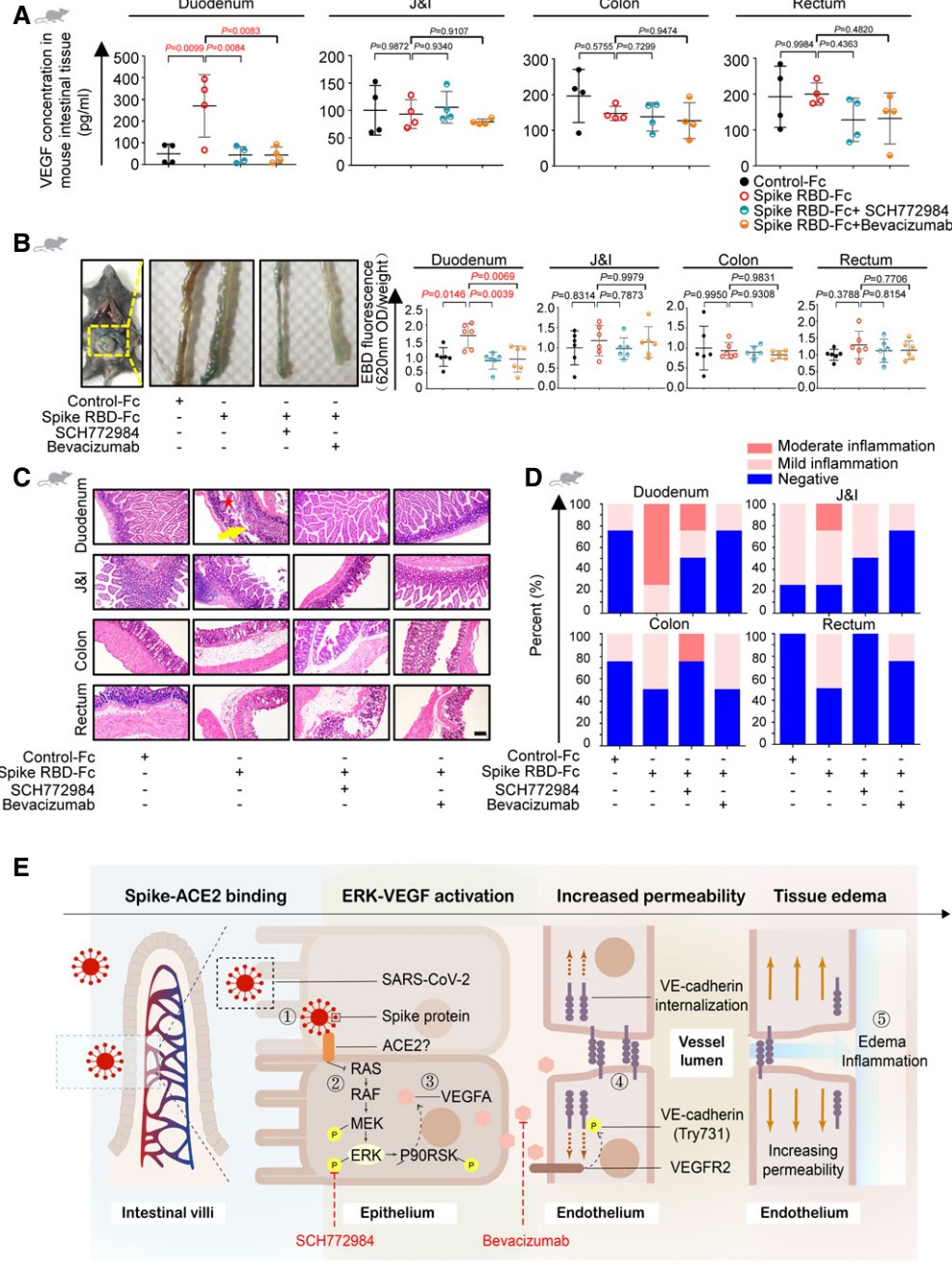

**Figure 5. Blockage of ERK/VEGF axis can alleviate SARS-CoV-2 spike RBD-induced hyperpermeability and intestinal inflammation.**

A    ELISA analysis of VEGF concentration in intestinal tissues of mice treated with ERK inhibitor or anti-VEGF antibody. SCH772984, ERK inhibitor; Bevacizumab, anti-VEGF antibody. For each group, $n = 4$. $n$, biologically independent samples (mice).

B    Evaluation of intestinal permeability in animals treated with SCH772984 or Bevacizumab by Evans Blue dye extravasation assay. For each group, $n = 6$. $n$, biologically independent samples (mice).

C, D  Representative H&E images (C) and the degrees of inflammation (D) of intestinal tissues from animals treated with SCH772984 or Bevacizumab. The inflammatory infiltrates were indicated by a yellow arrow. The edema area was indicated by a red star. Scale bar, 100 μm. For each group, $n = 4$. $n$, biologically independent samples (mice).

E    Model for the SARS-CoV-2 spike RBD-mediated vascular hyperpermeability and intestinal inflammation. ① Binding of spike RBD and receptors on enterocytes; ② Activation of the Ras-Raf-MEK-ERK pathway; ③ VEGF overproduction; ④ VEGF-triggered phosphorylation of Y731 and internalization of VE-cadherin; ⑤ Increase of endothelial permeability accompanied by interstitial edema and inflammation in the intestinal tissue.

Data information: All data are shown as mean ± SD. $P$ values are determined by one-way ANOVA.
Source data are available online for this figure.

# Materials and Methods

### Clinical samples

The clinical specimens of the COVID-19 patients ($n = 17$) including small intestine, colon, and rectum were collected using endoscopy in accordance with Chinese Center for Disease Control and Prevention (CDC) guidelines. The plasma samples at different time points were collected into heparinized plastic tubes from patients with COVID-19 (Test cohort: $n = 18$; Validation cohort: $n = 19$). The clinical information of the patients is shown in Table EV1. Ethical approval was obtained from the ethics committee of the Fifth Affiliated Hospital, Sun Yat-sen University (No. 2020K312-1). All patients involved in this study provided written informed consent and all experiments conformed to the principles set out in the WMA Declaration of Helsinki and the Department of Health and Human Services Belmont Report.

### Cell lines and cell culture

Human umbilical vein endothelial cells (HUVECs) were purchased from the ScienCell (Cat. No. 8000) and cultured in ECM (ScienCell, Cat. No. 1001) supplemented with 10% fetal bovine serum (FBS). Human colorectal adenocarcinoma cells (Caco-2) were purchased from the Guangzhou IGE Biotechnology (IGE2020032501), and NCM460 cells were gifted by Professor Kuang DM (Sun Yat-sen University) and cultured in Dulbecco's Modified Eagle's medium (DMEM, Invitrogen, Carlsbad, CA, USA) supplemented with 10% FBS. A murine endothelial cell line (C166) was obtained from the American Type Culture Collection (ATCC, Manassas, USA) and cultured in DMEM (Invitrogen, Carlsbad, CA, USA) supplemented with 10% FBS. The cells were cultured at 37°C in a humidified incubator with 5% $CO_2$. All the cell lines were tested negative for mycoplasma contamination.

### Immunohistochemical and Immunofluorescence staining

For tissue samples, immunohistochemistry was performed to determine the protein expression of ERK, pERK, VE-cad and pVE-cad (Y731) in duodenum, jejunum and ileum, colon and rectum. Briefly, formalin-fixed, paraffin-embedded tissue sections (4 μm) were blocked with 10% goat serum for 30 min at room temperature. Slides were incubated overnight at 4°C with the primary antibody for ERK (Cat. No. 4695, Cell Signaling Technology, 1:250), pERK (Cat. No.4370, Cell Signaling Technology, 1:400), VE-cad (Cat. No. AP73794-100, Abcepta, 1:200), and pVE-cad 731 (Cat. No. AP52553, Abcepta, 1:100). After rinsing with PBS, slides were incubated for 30 min at 37°C with HRP Polymer Conjugate (ZSGB-BIO). The quantitative analysis on the immunohistology was performed by two independent pathologists (J.H., Z.L.) using ImageJ software. The protein expression scores were based on optical density and produced a continuous value in the range of 0–300. Discrepancies were resolved by consensus.

For tissue immunofluorescence staining, sections were incubated with 10% goat serum in PBST for 1 h at room temperature and then incubated overnight at 4°C with primary antibodies (anti-ACE2, Santa Cruz, Cat. No. sc390851, 1:100; anti-SARS-CoV-2 spike, Sino Biological, Cat. No. 40150-R007, 1:500; Anti-CK, Cell Signaling Technology, Cat. No. 4545S,1:1,000; Anti-CD34, Thermo Fisher, Cat. No. MA1-10202,1:1,000; Anti-VEGF, Abcam, Ab52917, 1:2,000). The slides were incubated with secondary antibodies (Alexa Fluor®647-conjugated goat anti-rabbit IgG, bs-0296G-AF647, Bioss, 1:100; Dylight-550 Goat Anti-rabbit IgG secondary antibody, BA1135,1:200) for 1 h at room temperature followed by washing three times with PBST. Nuclei were then counterstained with 4', 6-diamidino-2-phenylindole (DAPI). Slides were imaged using a laser scanning confocal microscope (LSM880, Carl Zeiss Micro Imaging). The fluorescence intensity of spike protein was determined by ZEN Blue Lite 2.3 (Carl Zeiss Micro Imaging).

To detect the adherens junctions of endothelium, $2.5 \times 10^5$ HUVECs were seeded into 15-mm glass-bottom cell culture dish, incubated with the supernatant of cultured Caco-2 (treated with spike-Fc and IgG2b-Fc respectively) for 24 h, then fixed with 4% paraformaldehyde. Samples were stained with the following antibodies or fluorescent dyes: VE-cadherin (Cat. No. 44-1145G, Thermo Fisher, 1:1,000), Dylight-488 Goat Anti-mouse IgG secondary antibody (Cat. No. BA1126, BOSTER, 1:200) and Antifade Mounting Medium with DAPI (Cat. No. Ab104139, Abcam). For confocal microscopy, images were taken by Carl Zeiss LSM880 (Carl Zeiss MicroImaging, Inc.) and image processing was performed using ImageJ software (National Institutes of Health).

### Western blot

The tissues excised from mice (including control group and experimental group) were fixed and grinded immediately with liquid nitrogen. All the cells were lysed using RIPA buffer with a Protease/Phosphatase Inhibitors Cocktail (Beyotime Biotechnology, Shanghai, China). The proteins were separated in 5–20% SDS–PAGE gels (Genscript, Nanjing, China) and transferred onto PVDF (Sigma-Aldrich, USA) membranes for detection. The membranes were blocked using nonfat milk and incubated with primary antibodies against Actin (Cat. No. 4970S, Cell Signaling Technology, 1:1,000), ACE2 (Cat. No. NBP2-67692, Novus, 1:1,000), ZO-1 (Cat. No. AP21773-1, Abcepta, 1:1,000), VE-cadherin (Cat. No. AP73794, Abcepta, 1:1,000), p-VE-cad (Phospho-Tyr731) (Cat. No. 44-1145G, Thermo Fisher, 1:1,000), p-VE-cad (Phospho-Tyr658) (Cat. No. AF8206, Affinity, 1:1,000), RAS (Cat. No. 3965S, Cell Signaling Technology, 1:1,000), Phospho-Erk1/2 Pathway Sampler Kit (Cat. No. 9911T, Cell Signaling Technology, 1:1,000), p44/42 MAPK (Erk1/2) (Cat. No. 4695S, Cell Signaling Technology, 1:1,000), MEK1/2 (Cat. No. 8727T, Cell Signaling Technology, 1:1,000), and RSK1/RSK2/RSK3 (Cat. No. 9355T, Cell Signaling Technology, 1:1,000) at 4°C overnight. Then, the membranes were incubated with HRP Goat anti-rabbit IgG secondary antibody (Cat. No. E030130, EarthOx, 1:5,000) or HRP Goat anti-mouse IgG secondary antibody (Cat. No. E030110, EarthOx, 1:5,000). The protein bands were visualized with SUPERSIGNAL WEST PICO PLUS (Invitrogen) and detected by ChemiDoc/XPS⁺ (Bio-Rad). Image acquisition and quantitative analyses were performed with the Image Lab™ analyses system (Bio-Rad).

### Animal experiments

All animal procedures were performed in accordance with the Guidelines for the ethical review of laboratory animal welfare

(GB/T35892-2018) and approved by the Experimental Animal Ethics Committee of the Fifth Affiliated Hospital, Sun Yat-sen University. C57BL/6J male and female mice (age 7–8 weeks) were purchased from Beijing Weitong Lihua Experimental Animal Technical Co., Ltd. (Beijing, China). hACE2-B6J female mice (hACE2-All CDS-B6J, age 5–8 weeks) were purchased from Cyagen Biosciences. Animals were maintained at a constant temperature (26°C) under a light/dark cycle (14/10 h), and had ad libitum access to food and water. To generate the mouse model, mice were anesthetized with pentobarbital (30 mg/kg intraperitoneally (i.p.)) and induced with intestinal injury by rectal enema with 0.5 ml of 1% (v/v) acetic acid diluted in saline. Acetic acid solution was administered through a polyethylene catheter inserted into the rectum. The tip of catheter was positioned from 5 cm proximal to the anus verge.

To inhibit the spike-Fc-mediated intestinal inflammation, spike-Fc-challenged mice were administered with either ERK inhibitor (SCH772984, Cat. No. T6066-10, TargetMol, 50 mg/kg) or VEGF inhibitor (Bevacizumab, Cat. No. M6166, AbMole, 5 mg/kg) i.p. twice (one hour before and one hour after spike-Fc treatment) or mouse VEGF inhibitor (Mouse VEGF164 Antibody, Cat. No. AF-493-NA, R&D, 2.5 mg/kg). Six hours later, 200 µl TRITC-dextran (Invitrogen) were injected intravenously (i.v.). After specific time intervals, animals were euthanized and 2 ml of PBS was injected i.p. to recover the liquid in the peritoneal cavity. The solution was then spun down, and the supernatant was taken for quantification of fluorescence intensity. The intestinal tissue was subject to H&E staining. The plasma was collected before the treatment and after the injection daily for seven days.

### Study design and statistics of animal experiments

Mice were randomly allocated into experimental groups at the time of injection. Four investigators were involved in mice experiments: One investigator harvested and performed injection to the mouse, one investigator carried the surgery blindly (not knowing which condition was injected), and two investigators performed the quantification of phenotype. No statistical methods were used to estimate sample size. The only elimination criteria used for mouse studies were based on health.

### RNA isolation and qRT–PCR

Human tissues were collected by endoscopic intestinal biopsy. Mouse tissues were excised immediately after fixation in liquid nitrogen. Total RNA of tissues was isolated using TRIzol by the routine RNA extraction protocol used in our laboratory. Total RNA of cells was isolated by E.Z.N.A.® Total RNA Kit I. The isolated RNA was reverse transcribed into cDNA (Vazyme, Nanjing, China). Quantitative reverse transcription PCR (qRT–PCR) was performed using Real-Time PCR system (Bio-Rad, America) and ChamQ Universal SYBR qPCR Master Mix (Vazyme, Nanjing, China). The primers were designed and synthesized by Guangzhou IGE Biotechnology (Guangzhou, China). GAPDH was used as internal controls, and all reactions were performed in triplicate. Relative RNA expression was calculated using the $2^{-\Delta\Delta Ct}$ method. Primers used in this study were listed in table below.

| Name | Forward primer | Reverse primer |
| --- | --- | --- |
| VEGF | GCACATAGAGAGAATGAGCTTCC | CTCCGCTCTGAACAAGGCT |
| IL-1α | AGCTTGACGGCACCCTCGCA | CGGAGAGCTTCGTGGCTGTGGA |
| IL-8 | CAGCTGCCTTAACCCCATCA | CTTGAGAAGTCCATGGCGAAA |
| IL-10 | ATCGATTTCTCCCCTGTGAA | TGTCAAATTCATTCATGGCCT |
| TNF-α | CCACCACGCTCTTCTGTCTAC | AGGGTCTGGGCCATAGAACT |
| VEGF-A | GCCTTGCCTTGCTGCTCTAC | TGATTCTGCCCTCCTCCTTCTG |
| VEGF-B | AGGACAGAGTTGGAAGAGGAG | AGGAAGAGCCAGTTGTAAGATG |
| VEGF-C | ATGTTTTCCTCGGATGCTGGA | CATTGGCTGGGGAAGAGTTT |
| VEGF-D | GTATGGACTCTCGCTCAGCAT | AGGCTCTCTTCATTGCAACAG |
| KRAS | TCCCAGGTGCGGGAGAG | TTAGCTGTATCGTCAAGGCACT |
| HRAS | TTCTACACGTTGGTGCGTGA | GGGAGTCCCCCTCACCTG |
| NRAS | CTCACTTGGCTGTCTGACCA | GATGAAAAACCTGGGGTGGC |
| RAF1 | CCGTGTTTTCTTGCCGAACA | TGAACACTGCACAGCACTCT |
| MAPK | TCCTTTGAGCCGTTTGGAGG | TACATACTGCCGCAGGTCAC |
| MAP2K7 | GGCAACAGGACAGTTTCCCT | AGGCAGTCTTTGACGAAGGA |
| P90RSK | AAGAGTAACGGGGCCCTCTG | GCCCTCATCCTTTCTGGGAC |

### RNA interference

Caco-2 cells were transfected for 48 h with a negative control siRNA and ACE2 siRNA (Cat. No. sipack_1999hACE2, RIBOBIO) using Lipofectamine LTX (Invitrogen, Thermo Fisher Scientific) and then were lysed using RIPA buffer with a Protease/Phosphatase Inhibitors Cocktail to prepare for Western blot.

### Permeability assay

Caco-2 cells ($5 \times 10^5$) were treated with control IgG-Fc (0.25 mg/ml), spike RBD-Fc (0.25 mg/ml) or spike RBD-Fc combined with SCH772984 (1 µM) (Cat. No. T6066-10, TargetMol), or Bevacizumab (25 µg/ml) (Cat. No. M6166, Abmole), respectively, for 24 h. Then, the supernatant of cultured Caco-2 cells was filtered by 0.22-µm strainer. HUVECs ($1 \times 10^5$) monolayer was grown in a transwell insert in 24-well transwell chambers (0.4 µm pore size; Corning) overnight. After incubation for 24 h until reaching confluence, the bottom of transwell chambers was changed with the filtered supernatant of cultured Caco-2 and co-cultured with HUVECs for 12 h. To detect the HUVEC permeability in vitro, the medium of upper chamber was removed and Tetramethylrhodamine isothiocyanate–Dextran (TRITC-Dextran) (Cat. No. T1162, Sigma) (2 mg/ml) was added to the upper well. After 3 h, the medium with TRITC-Dextran of bottom well were collected and measured for the fluorescence at excitation and emission wavelengths of 540 and 590 nm, respectively, using a microplate reader.

### ELISA

The concentration of VEGF generated from cell supernatant, tissues, and plasma was detected by ELISA. VEGF concentration in human cell culture supernatant and COVID-19 patient plasma samples ($n = 19$) in validation cohort was determined by the Quantikine ELISA Human VEGF Immunoassay (Cat. No. DVE00, R&D) according to the manufacturer's instructions. The detailed

information of the 19 patients are shown in Table EV1. For mouse samples, the VEGF levels from tissue homogenates (prepared within cold PBS using an electric homogenizer) or plasma were determined using Mouse VEGF Simplestep ELISA Kit (Cat. No. ab209882, Abcam) following the manufacturer's instructions.

### Cytokine measurement

Vascular permeability related factors, including IL-1α, IL-1β, TNF-α, IFN-γ, and VEGF, were measured using Luminex Assay Human XL Cytokine Discovery Premixed Kit (R&D) and the Luminex® 200™ system (Millipore) according to the manufacturer's instructions.

### Statistics

Statistical analyses were performed using GraphPad PRISM software v8.4.2 (GraphPad Software Inc, San Diego, CA, USA). As indicated, statistical analysis was performed by calculating mean $\pm$ SD. Student's $t$-test was used when only two groups were analyzed. For comparisons between multiple groups, homogeneity variance was first tested, and the one-way ANOVA was applied when variance was homogeneous, or non-parametric independent sample $t$-test (the Mann–Whitney test) was used for inhomogeneity. Paired Student's $t$-test or one-way ANOVA was used to analyze the quantitative Western blot as indicated in figure legends. Pearson's correlation was performed to analyze the correlation between clinical parameters. Two-tailed test was used for all $t$-tests, and $P < 0.05$ was considered statistically significant.

### Ethics approval

All animal experiments were approved by the Experimental Animal Ethics Committee of the Fifth Affiliated Hospital, Sun Yat-sen University, and performed in accordance with the Guidelines for the ethical review of laboratory animal welfare (GB/T35892-2018, Number 00108/11210). Studies related to human samples were approved by the Medical Ethics Committee of the Fifth Affiliated Hospital, Sun Yat-sen University.

## Data availability

The RNA-seq data of patients have been deposited to the GSA-human database with accession number HRA002115 (https://ngdc.cncb.ac.cn/gsa-human/s/geaE1XIf) in accordance to ethical obligations to patients and ethical approval.

**Expanded View** for this article is available online.

### Acknowledgements

The authors would like to thank Professor Man Li (Fifth Affiliated Hospital, Sun Yat-sen University) for statistical analysis and Professors Pei-Hui Wang (Shandong University), Musheng Zeng, Song Gao, Kai Deng, Zhixian Yan, Shoudeng Chen, and Dongming Kuang (all from Sun Yat-sen University) for providing experimental materials.

This work was supported by grants from The National Natural Science Foundation of China (81872113, 81870411, 82172241), The National Key

**The paper explained**

**Problem**

COVID-19 patients with gastrointestinal (GI) symptoms tend to develop severe disease, but the underlying mechanism is unclear. VEGF is upregulated in the blood of COVID-19 patients, yet its association with the development of GI symptoms has not been explored.

**Results**

VEGF level correlated with intestinal edema, GI symptoms, and disease progression in COVID-19 patients. In an animal model mimicking intestinal inflammation upon stimulation with SARS-CoV-2 spike protein, we found that VEGF was over-produced in the duodenum prior to its ascent in the circulation, which led to hyperpermeability and systemic inflammation. Cell experiments demonstrated that the SARS-CoV-2 spike protein activated the Ras-Raf-MEK-ERK-VEGF pathway in enterocytes, which promoted VE-cad-mediated vascular permeability. Blocking the ERK/VEGF axis reversed hyperpermeability and alleviated intestinal inflammation stimulated by SARS-CoV-2 spike protein both *in vitro* and *in vivo*.

**Impact**

This study identifies a possible route of SARS-CoV-2 spike-induced VEGF production in the GI tract, which leads to vascular permeability and inflammation. Mechanistically, it uncovers the spike-activated ERK/VEGF pathway in enterocytes. Translationally, it provides potential targets for the treatment of GI symptoms, harnessing the disease progression of COVID-19.

Research and Development Plan for the Emergency Management of Novel Coronavirus Pneumonia (2020YFC0842400), The Special Project of Guangdong Science and Technology Department (2020B111113001), The Science and Technology Program of Guangzhou (202008030001), and The Novel Coronavirus Epidemic Prevention and Control Emergency Project of Zhuhai City (ZH22036302200005PWC).

## Author contributions

**Fa-Min Zeng:** Data curation; Formal analysis; Funding acquisition; Investigation; Visualization; Writing—original draft. **Ying-wen Li:** Data curation; Formal analysis; Investigation; Visualization; Writing—original draft. **Zhao-hua Deng:** Data curation; Investigation. **Jian-zhong He:** Data curation; Formal analysis; Investigation. **Wei Li:** Methodology. **Lijie Wang:** Methodology. **Ting Lyu:** Methodology. **Zhanyu Li:** Methodology. **Chaoming Mei:** Methodology. **Meiling Yang:** Investigation; Methodology. **Yingying Dong:** Investigation; Methodology. **Guan-Min Jiang:** Resources. **Xiaofeng Li:** Investigation; Methodology. **Xi Huang:** Resources; Investigation. **Fei Xiao:** Conceptualization; Resources; Supervision; Validation; Methodology. **Ye Liu:** Resources; Supervision; Validation. **Hong Shan:** Conceptualization; Resources; Supervision; Funding acquisition; Validation. **Huanhuan He:** Conceptualization; Data curation; Formal analysis; Supervision; Funding acquisition; Validation; Visualization; Project administration; Writing—review & editing.

In addition to the CRediT author contributions listed above, the contributions in detail are:

HH, HS, YL, and FX are co-corresponding authors and take equal full responsibilities for all contents of paper. HH appears as last senior authors because she supervised the study and were responsible for the critical appraisal of the final manuscript. HS, YL, and FX are responsible for providing clinical samples and information, and administrative support. F-mZ, Y-wL, Z-hD, and J-zH are co-first authors and were responsible for collection of data and writing the

initial draft of the manuscript. F-mZ appears as first author because she directed the work, designed experiments and organized the manuscript. F-MZ, Y-wL, Z-hD, J-zH, WL, LW, TL, ZL, CM, MY, and YD performed the experiments. F-mZ and J-zH analyzed data. XH, XL, and G-MJ provided clinical samples. All authors have read and approved the manuscript.

## Disclosure and competing interests statement

The authors declare that they have no conflict of interest.

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
