## [Review Process File · EMBO Molecular Medicine]

SARS-CoV-2 Spike Spurs Intestinal Inflammation via VEGF Production in Enterocytes

Fa-min Zeng, Ying-wen Li, Zhao-hua Deng, Jian-zhong He, Wei Li, Lijie Wang, Ting Lyu, Zhanyu Li, Chaoming Mei, Meiling Yang, Ying-ying Dong, Guan-Min Jiang, Xiaofeng Li, Xi Huang, Fei Xiao, Ye Liu, Hong Shan, and Huanhuan He

DOI: [10.15252/emmm.202114844](https://doi.org/10.15252/emmm.202114844)

Corresponding authors: Huanhuan He (hehh23@mail.sysu.edu.cn), Hong Shan (shanhong@mail.sysu.edu.cn), Ye Liu (ly77219@163.com), Fei Xiao (xiaof35@mail.sysu.edu.cn)

Review Timeline:

Submission Date:	29th Sep 21
Editorial Correspondence:	27th Oct 21
Author Correspondence:	28th Oct 21
Editorial Decision:	2nd Nov 21
Revision Received:	24th Feb 22
Editorial Decision:	1st Mar 22
Revision Received:	18th Mar 22
Accepted:	24th Mar 22

Editor: Lise Roth

Transaction Report:

Dear Prof. He,

I am contacting you regarding your manuscript EMM-14844 currently under review at EMBO Molecular Medicine. Please accept my apologies for the delay in getting back to you, as we are still waiting for the report from one referee.

We already received two reports, and a main issue that was raised comes from a confusion regarding the model you used. In particular, one referee stated:

"The spike protein of SARS-CoV2 does not bind well to murine ACE2; from reading the materials and methods it is not clear how this was addressed experimentally in this paper. Did the studies involve transgenic hACE2 mice (does not seem to be the case from is written in the methods section)? Did the studies use mouse adapted spike RBD sequences (for example, see this paper <https://doi.org/10.1016/j.ebiom.2021.103381>)? Otherwise, it is hard to make sense of the story."

As this is a crucial point for our decision making, we would appreciate if you could clarify already now whether you used murine adapted Spike sequences or hACE2 transgenic mice.

Looking forward to your answer,

With kind regards,

Lise Roth

Dear Dr. Roth,

We appreciate the concern you and the referee raised, which was very reasonable. We had carefully considered this issue during experimental plan and have included below the evidence from our manuscript and previous reports to support our statement.

In short, the current study did not use transgenic hACE2 mice or mouse adapted Spike RBD sequences, for the reason that we have plenty of solid evidence to support **the usage of Spike RBD for establishing murine models**. In addition, existing reports and our experimental confirmation support that **murine ACE2 binds to SARS-CoV-2 Spike RBD although not to the extent that human ACE2 does**. However, binding does not guarantee infection since infection involves many steps such as cleavage of Spike glycoprotein and entry of virus. Therefore, **what we have demonstrated in this work is how Spike RBD could trigger intestinal inflammation and downstream molecular cascade**. This mechanism describes only one part of the consequences of SARS-CoV-2 Spike's early contact with enterocytes, as what happens upon viral entry is beyond the scope of this work and is not suitable to study by the current model.

Specific evidence and views that support our model are as follows:

1. In our work, we proved the affinity of mACE2 and Spike RBD.

1) Binding of Spike RBD to mACE2 was tested *in vitro* by co-immunoprecipitation (**Appendix Figure 2B**);

2) Colocalization of Spike RBD and mACE2 was confirmed in the intestinal tissue from COVID-19 patients by immunofluorescence (**Appendix Figure 2C, D**).

These results above suggest sufficient affinity of mACE2 and Spike RBD. We are currently examining their binding affinity using surface plasmon resonance (SPR) assay and the result will be uploaded in our revised manuscript.

2. Existing reports supporting mACE2 binds to Spike of either SARS-CoV-2 or SARS-CoV.

1) A crucial role of angiotensin converting enzyme 2 (ACE2) in SARS coronavirus-induced lung injury. (Nat Med. 2005;11(8):875-9. doi: 10.1038/nm1267. **Figure 2 c, d**)

The authors intraperitoneally injected Spike-Fc Spike protein to wild-type mice and found it aggravated murine lung injury. The authors also confirmed the binding of mACE2 with SARS-CoV Spike using Co-IP and FACS.

2) Receptor-binding domain of SARS-CoV-2 spike protein efficiently inhibits SARS-CoV-2 infection and attachment to mouse lung. (Int J Biol Sci. 2021;17(14):3786-3794. doi: 10.7150/ijbs.61320. **Figure 4**)

The authors intranasally injected SARS-CoV-2 to wild-type mice and showed that the attachment of SARS-CoV-2 to mouse lungs was detectable using qRT-PCR. They also detected the binding of mACE2 and Spike in previous work (see **Point 3**) below).

3) Broad and Differential Animal Angiotensin-Converting Enzyme 2 Receptor Usage by SARS-CoV-2. J Virol. 2020;94(18):e00940-20. doi: 10.1128/JVI.00940-20. **Figure 3 c**)

In this work, the binding of mACE2 and Spike was determined by Co-IP and luciferase assay.

4) The *Rhinolophus affinis* bat ACE2 and multiple animal orthologs are functional receptors for bat coronavirus RaTG13 and SARS-CoV-2. (Sci Bull (Beijing). 2021;66(12):1215-1227. doi: 10.1016/j.scib.2021.01.011. **Figure 3 b,c**)

Similar results demonstrated that Spike RBD bound to mACE2 *in vitro*.

5) Danshensu alleviates pseudo-typed SARS-CoV-2 induced mouse acute lung inflammation. (Acta Pharmacol Sin. 2021;1-10. doi: 10.1038/s41401-021-00714-4. **Figure 2a; Figure 5b**)

SARS-CoV-2 Spike induced acute lung inflammation in adult BALB/c mice. In addition, SARS-CoV-2 Spike bound to murine lung tissues and inhibited the mRNA level of mACE2.

In summary, existing reports and our data propelled us to believe that SARS-CoV-2 Spike RBD DOES bind to murine ACE2, although less efficiently than it binds to human ACE2 as indicated by Li W et al. in *Efficient Replication of Severe Acute Respiratory Syndrome Coronavirus in Mouse Cells Is Limited by Murine Angiotensin-Converting Enzyme 2*. J Virol. 2004;78(20):11429-33. doi: 10.1128/JVI.78.20.11429-11433.2004. **Figure 1 A-C**). It is widely appreciated that mice are immune to SARS-CoV-2 infection, yet it may be due to other reasons rather than the binding issue. Since infection involves a series of processes including binding of Spike RBD to ACE2, cleavage of Spike glycoprotein and entry of virus, etc., binding itself does not guarantee infection.

In conclusion, strong evidence supports that SARS-CoV-2 Spike RBD can bind to murine ACE2 and our animal model reveals the downstream effects of this binding in vivo. If given the chance to revise, we will provide the SPR data to examine the affinity of mACE2 to Spike RBD and replicate our findings in transgenic hACE2 mice.

*The data reported by previous studies have been attached.

1) A crucial role of angiotensin converting enzyme 2 (ACE2) in SARS coronavirus-induced lung injury. (Nat Med. 2005;11(8):875-9. doi: 10.1038/nm1267. **Figure 2 c,d**)

2) Receptor-binding domain of SARS-CoV-2 spike protein efficiently inhibits SARS-CoV-2 infection and attachment to mouse lung. (Int J Biol Sci. 2021;17(14):3786-3794. doi: 10.7150/ijbs.61320. **Figure 4**)

3) Broad and Differential Animal Angiotensin-Converting Enzyme 2 Receptor Usage by SARS-CoV-2. J Virol. 2020;94(18):e00940-20. doi: 10.1128/JVI.00940-20. **Figure 3 c)**

4) The *Rhinolophus affinis* bat ACE2 and multiple animal orthologs are functional receptors for bat coronavirus RaTG13 and SARS-CoV-2. (Sci Bull (Beijing). 2021;66(12):1215-1227. doi: 10.1016/j.scib.2021.01.011. **Figure 3 b, c)**

5) Danshensu alleviates pseudo-typed SARS-CoV-2 induced mouse acute lung inflammation. (Acta Pharmacol Sin. 2021;1-10. doi: 10.1038/s41401-021-00714-4. **Figure 2a; Figure 5b)**

6) Efficient Replication of Severe Acute Respiratory Syndrome Coronavirus in Mouse Cells Is Limited by Murine Angiotensin-Converting Enzyme 2. *J Virol.* 2004;78(20):11429-33. doi: 10.1128/JVI.78.20.11429-11433.2004. **Figure 1 A-C).**

2nd Nov 2021

Dear Prof. He,

Thank you for the submission of your manuscript to EMBO Molecular Medicine, and please accept my apologies for the delay in getting back to you, which is due to the fact that we were waiting for the report from the third referee. As we have not received it yet, we prefer to make a decision now based on both referees #1 and #2's reports in order to avoid further delay in the process. Should referee #3 provide a report shortly, we will send it to you, with the understanding that we would not ask you for further-reaching experiments in addition to the ones required in the enclosed reports.

As mentioned in a previous exchange, one of the main issues raised in the reports related to the adequacy of the model system used. You kindly provided further explanation on that point, which we forwarded to referees #1 and #2.

Referee #2 stated:

"I think there is a reason people use hACE2 transgenic mice and mouse adapted SARS-CoV2 strains. Moreover, I believe it is incumbent upon the authors to convince us that the model is indeed relevant, and there are ways to do this.

* If the claim is that there is enough contact of Spike to murine ACE2 to reach relevant conclusions, that may be ok but there has to be additional confirmation. One simple thing they could do is confirm with a murine ACE2 knockout that the responses they see to Spike-RBD-Fc are ACE2-dependent and not some artifact of other activities of their recombinant protein. I did not find an ACE2 k/o model right away, but again, the onus is on them to address the concern of model suitability.

* I would also think that doing a key confirmatory experiment with the hACE2 transgenic would be important: I don't think that they need to repeat the whole paper, but prove that the story is reliable, even though the ACE2 receptor and spike protein are not perfectly matched in the first set of experiments. On a quick search, there are lots of hACE2 transgenic mice available from Jax labs."

Upon discussion with my colleagues, we agree that addressing these points as well as the other referees' concerns will be necessary for further considering the manuscript in our journal. Acceptance of the manuscript will entail a second round of review. EMBO Molecular Medicine encourages a single round of revision only and therefore, acceptance or rejection of the manuscript will depend on the completeness of your responses included in the next, final version of the manuscript. For this reason, and to save you from any frustrations in the end, I would strongly advise against returning an incomplete revision.

We require:

3) A .docx formatted letter INCLUDING the reviewers' reports and your detailed point-by-point responses to their comments. As part of the EMBO Press transparent editorial process, the point-by-point response is part of the Review Process File (RPF), which will be published alongside your paper.

4) A complete author checklist, which you can download from our author guidelines (<https://www.embopress.org/page/journal/17574684/authorguide#submissionofrevisions>). Please insert information in the checklist that is also reflected in the manuscript. The completed author checklist will also be part of the RPF.

5) Please note that all corresponding authors are required to supply an ORCID ID for their name upon submission of a revised manuscript. We note that you currently have together with you, a total of 4 co-corresponding authors. Is that correct? Do you confirm equal contribution of these 4 people, able to take full responsibility for the paper and its content? While there is no limit per se to the number of co-corresponding authors, 3 is rare, 4 even more so, and may not reflect as intended to the community.

6) It is mandatory to include a 'Data Availability' section after the Materials and Methods. Before submitting your revision, primary datasets produced in this study need to be deposited in an appropriate public database, and the accession numbers and database listed under 'Data Availability'. Please remember to provide a reviewer password if the datasets are not yet public (see <https://www.embopress.org/page/journal/17574684/authorguide#dataavailability>).

7) For data quantification: please specify the name of the statistical test used to generate error bars and P values, the number (n) of independent experiments (specify technical or biological replicates) underlying each data point and the test used to calculate p-values in each figure legend. The figure legends should contain a basic description of n, P and the test applied. Graphs must include a description of the bars and the error bars (s.d., s.e.m.).

8) We would also encourage you to include the source data for figure panels that show essential data. Numerical data should be provided as individual .xls or .csv files (including a tab describing the data). For blots or microscopy, uncropped images should be submitted (using a zip archive if multiple images need to be supplied for one panel). Additional information on source data and instruction on how to label the files are available at

9) Our journal encourages inclusion of *data citations in the reference list* to directly cite datasets that were re-used and obtained from public databases. Data citations in the article text are distinct from normal bibliographical citations and should directly link to the database records from which the data can be accessed. In the main text, data citations are formatted as follows: "Data ref: Smith et al, 2001" or "Data ref: NCBI Sequence Read Archive PRJNA342805, 2017". In the Reference list, data citations must be labeled with "[DATASET]". A data reference must provide the database name, accession number/identifiers and a resolvable link to the landing page from which the data can be accessed at the end of the reference. Further instructions are available at

13) Every published paper now includes a 'Synopsis' to further enhance discoverability. Synopses are displayed on the journal webpage and are freely accessible to all readers. They include a short stand first (maximum of 300 characters, including space) as well as 2-5 one-sentences bullet points that summarizes the paper. Please write the bullet points to summarize the key NEW findings. They should be designed to be complementary to the abstract - i.e. not repeat the same text. We encourage inclusion of key acronyms and quantitative information (maximum of 30 words / bullet point). Please use the passive voice. Please attach these in a separate file or send them by email, we will incorporate them accordingly.

14) As part of the EMBO Publications transparent editorial process initiative (see our Editorial at <http://embomolmed.embopress.org/content/2/9/329>), EMBO Molecular Medicine will publish online a Review Process File (RPF) to accompany accepted manuscripts.

In the event of acceptance, this file will be published in conjunction with your paper and will include the anonymous referee reports, your point-by-point response and all pertinent correspondence relating to the manuscript. Let us know whether you agree with the publication of the RPF and as here, if you want to remove or not any figures from it prior to publication. Please note that the Authors checklist will be published at the end of the RPF.

EMBO Molecular Medicine has a "scooping protection" policy, whereby similar findings that are published by others during

review or revision are not a criterion for rejection. Should you decide to submit a revised version, I do ask that you get in touch after three months if you have not completed it, to update us on the status.

I look forward to seeing a revised form of your manuscript as soon as possible. Use this link to login to the manuscript system and submit your revision: Link Not Available

Yours sincerely,

Lise Roth

Lise Roth, Ph.D
Editor
EMBO Molecular Medicine

***** Reviewer's comments *****

Referee #1 (Comments on Novelty/Model System for Author):

In this manuscript, the authors found that VEGF was high in the duodenum of COVID19 patients. By combining in vitro cellular and mouse models, they then described that VEGF might source from enterocytes upon ERK activation. This might ultimately lead to hyper-permeability, and therefore inflammation and tissue edema. The exploration of human patients and liquid biopsies is also viewed as commendable. However, the manuscript might fall short in terms of mechanistic insights.

Referee #1 (Remarks for Author):

In this manuscript, the authors found that VEGF was high in the duodenum of COVID19 patients. By combining in vitro cellular and mouse models, they then described that VEGF might source from enterocytes upon ERK activation. This might ultimately lead to hyper-permeability, and therefore inflammation and tissue edema. The exploration of human patients and liquid biopsies is also viewed as commendable. However, the manuscript might fall short in terms of mechanistic insights.

- VEGF may come as multiple forms; there is no information on the nature of the secreted VEGF. VEGF-A? -165, -121? Others? This may have strong patho-physiological implications and required further investigation. Levels of VEGFRs should also be documented.
- How do the authors reconcile the local production of VEGF suggested by IHC to the systemic increased in the plasma?
- In the Fig 4, the authors conclude on increased on pERK, while ERK is also upregulated. Quantification of phosphorylated forms should be done on corresponding total protein (ie pERK/ERK, pMEK/MEK and p-P90RSK/P90RSK). In addition, total MEK and total P90RSK are missing. One would ask whether this increased is at the RNA level?
- The authors show by western-blot a decreased in total VE-cadherin. What are the molecular mechanisms behind? This cannot be due to increased internalization, as suggested. The authors should show by WB pVE-cad/VE-cad in conditions similar to panel E.
- Moreover, the authors should check the involvement of VEGFRs in the endothelial cells challenged with Caco2-supernatant.
- The authors exclude a direct effect from spike protein on endothelial cells, based on measuring VEGF level on HUVEC. This model would merit to be challenged to mimic the intestine environment endothelium. One cannot exclude either an ACE-dependent modulation of the endothelial integrity in the in vivo experiment. This has to be challenged. It is not either clear how effective is bevacizumab in mice.

Referee #2 (Comments on Novelty/Model System for Author):

The problems with the models are two:

- The authors need to clarify whether they used murine adapted Spike sequences or hACE2 transgenic mice (see comment below).
- The entire story is based on Spike-RBD-Fc protein treatments in vivo and the story lacks a single experiment with live SARS-CoV2. I suggest that a confirmatory experiment at the end, using SARS-CoV2 and Bevacizumab, would strengthen the story.

Referee #2 (Remarks for Author):

In this manuscript, we are introduced to the idea that Spike protein acting on ACE2 in enterocyte can mediate ERK activation, VEGF overexpression, and ultimately, increased gut permeability and worse clinical course. Overall, the story has many strong points and its potential translational value is significant. In my assessment, there are 3 major concerns related to the paper that would need to be addressed first in order for this story to be ready for publication:

- 1) The clinical data in Figure 1 is based on a relatively small number of patients. While some aspects of the data are certainly hard to expand (particularly the GI tract biopsies), the serologic analysis of VEGF levels and GI symptoms or disease progression could be easily expanded, or recapitulated in an independent and separate cohort of patients. I believe this to be critical to confirm the validity and reproducibility of this key aspect of the study. In addition, this would allow for other analyses, like longitudinal evaluation of VEGF during disease progression, and correlation of VEGF with other serologic markers of disease severity (CRP, ferritin, d-dimers).
- 2) The spike protein of SARS-CoV2 does not bind well to murine ACE2; from reading the materials and methods it is not clear how this was addressed experimentally in this paper. Did the studies involve transgenic hACE2 mice (does not seem to be the case from is written in the methods section)? Did the studies use mouse adapted spike RBD sequences (for example, see this paper <https://doi.org/10.1016/j.ebiom.2021.103381>)? Otherwise, it is hard to make sense of the story.
- 3) Many of the changes in the ERK and other pathways that are shown by western blot are subtle; at times, the quantification shown in bar graphs is not representative of the western blots shown (see for example pMEK in Fig 3H). Moreover, the authors interpret Fig S3D and E as evidence that Spike RBD mediates its effects through ACE2 and not through TMPRSS2, using siRNA experiments. However, the experiments seem to be simply an analysis of these kinases after siRNA of these receptors and do not include Spike RBD treatment, so they simply speak to the impact of those receptors on these pathways. Overall, these data need to be significantly strengthened - showing data points in the bar graphs would help, also it is noted that the pERK changes are most dramatic in duodenal tissue in vivo (Fig S4A) and this seems to be a more robust and physiologically significant model than cells in culture. The authors are encouraged to use this model for many of the data shown in 3C and 3H. In addition, as alluded to already, the siRNA experiments need Spike-RBD treatment (Figs S3D, E).
- 4) The experiments using Bevacizumab in vivo looking at gut permeability are quite nice, and one of the stronger parts of the paper. The story would be a lot stronger if the same experiment was performed in mice with COVID infection (either using a mouse adapted strain or ACE2 transgenic mice).

Other comments:

- 1) The following statement did not seem to correspond to the data that is being referenced: "Notably, interstitial edema was significantly correlated with disease type, acid reflux, total bilirubin, glutamic-pyruvic transaminase (ALT) and aspartate aminotransferase (AST) (Table 1 and Appendix Table S1), indicating vascular permeability change as a crucial factor in disease progression." Table 1 shows the correlation coefficients of various clinical parameters and VEGF levels and Appendix Table S1 shows the clinical characteristics of the cohort. Please clarify where the data actually is.
- 2) The Y axis of Figure 2F reads VEGF levels, but other cytokines are actually shown. Please correct.
- 3) ACE2 expression in HUVEC and Caco-2 (Fig 3A, B) needs to be shown in order to interpret this data.

POINT-TO-POINT RESPONSE TO REVIEWERS

We would like to thank the Reviewers and the Editors for their efforts in evaluating this work and for their constructive comments and criticisms. The author team has discussed the review and carefully conducted a series of experiments to address the points raised. This has resulted in the incorporation of **12** new or revised panels in main figures, **23** panels in EV figures, **3** new or revised tables, **1** revised EV table. New or revised panels are indicated in red below. Several editorial changes were also made in the main manuscript. The outcome of the review process has been a stronger study and therefore we are grateful!

Reviewer #1

1. VEGF may come as multiple forms; there is no information on the nature of the secreted VEGF. VEGF-A? -165, -121? Others? This may have strong patho-physiological implications and required further investigation. Levels of VEGFRs should also be documented.

Response: Thank you for pointing this out as ascertaining the isoforms of VEGF and the type of VEGF receptors will have strong clinical significance. We have made a comprehensive investigation and the results are as follows:

- (1) The ELISA kit we use (Cat. No. ab222510, Abcam) for the detection of VEGF in the serum of patients is designed to detect both human VEGF-165 and VEGF-121. Therefore, we were unable to distinguish the individual isoforms. Yet, based on previous research, no significant difference was observed between the effects of VEGF-165 and VEGF-121 on vascular permeability (Xu et al. JBC. 2011).
- (2) For VEGF family members, we attempted to examine VEGF-A-D levels in both co-culture systems and clinical samples. In spike-treated Caco-2, an up-regulation of VEGF-A, B and D was detected compared to the control group by qPCR (**Fig EV4A**). This is partially consistent with the data from the intestinal tissues of COVID-19 patients, in which VEGF-A was also significantly elevated but the levels of VEGF-B and C were not altered analyzed by RNA-seq (**Fig 2A** and **Fig EV1F**). Unfortunately, the VEGF-D result was not available in the RNA-seq data.
- (3) For VEGF receptors, the mRNA levels of VEGFR1, R2 and R3 in the intestinal tissues of COVID-19 patients showed no significant difference compared to those in healthy controls, thus we did not explore further (**Fig EV1G**).

2. How do the authors reconcile the local production of VEGF suggested by IHC to the systemic increased in the plasma?

Response: Excellent point. This observation is definitely worth discussing. Based on the results from the animal model and the clinical samples, we speculate that the systemic increase of VEGF might be initiated and contributed by the local production of VEGF with evidence as follows:

- (1) Firstly, in COVID-19 patients, we detected a higher VEGF mRNA level in the intestinal tissues (Fig 2A, 3C and Fig EV4B) and an elevated level of VEGF protein in the plasma (Fig 1C, D and Fig EV1C, D), which separately confirmed local production and systemic increase of VEGF.
- (2) More importantly, we performed a seven-day stimulation of spike RBD in the animal model, in which serum VEGF was not elevated initially, but markedly increased after seven days of continual treatment of spike RBD (Fig 2G and Fig EV3E). Since the level of murine VEGF in intestine raised even upon short contact with spike RBD (Fig 2F; Fig EV3C and D), these observations point to the contribution of local induction of VEGF to the systemic VEGF level. However, there might be inflammation or tissue damage in other organs that could also contribute to the VEGF in circulation in COVID-19 patients and we have discussed these points in the Discussion section (P12, L8-12).

3. In the Fig 4, the authors conclude on increased on pERK, while ERK is also upregulated. Quantification of phosphorylated forms should be done on corresponding total protein (ie pERK/ERK, pMEK/MEK and p-P90RSK/P90RSK). In addition, total MEK and total P90RSK are missing. One would ask whether this increase is at the RNA level?

Response: Good point. We have included total MEK and total P90RSK and carried out an extensive analysis of the mRNA and protein levels of the molecules on the ERK pathway (Fig 3I; Fig EV4E). These results reveal a classical activation of the ERK pathway.

4. The authors show by western-blot a decreased in total VE-cadherin. What are the molecular mechanisms behind? This cannot be due to increased internalization, as suggested. The authors should show by WB pVE-cad/VE-cad in conditions similar to panel E.

Response: Thank you for pointing this out. We have added WB of pVE-cad/VE-cad in HUVECs in various conditions and confirmed that spike RBD could reduce the total VE-cad while increasing pVE-cad (Fig EV5A). As for the mechanism of VEGF-mediated regulation of VE-cad, ample researches have been done and have demonstrated that the activation of VEGFR2 and AKT, NOS, ERK-dependent pathways. (Wettschureck et al. *Physiol Rev.* 2019; Lal et al. *Microvasc Res.* 2001). We have also confirmed the elevated pVEGFR2 in HUVECs co-cultured with

Caco2-supernatant. However, since this is beyond the investigation of this work, we have attached the data below for your reference (Unpublished Fig 1).

5. Moreover, the authors should check the involvement of VEGFRs in the endothelial cells challenged with Caco2-supernatant.

Response: We have examined the level of pVEGFR2 by Western blot (Unpublished Fig 1). The mRNAs of VEGFR1-3 detected from intestinal tissues of COVID-19 patients showed no difference across different groups (Fig EV1G), therefore we did not examine the levels of VEGFR1 and VEGFR3 in endothelial cells.

6. The authors exclude a direct effect from spike protein on endothelial cells, based on measuring VEGF level on HUVEC. This model would merit to be challenged to mimic the intestine environment endothelium. One cannot exclude either an ACE-dependent modulation of the endothelial integrity in the in vivo experiment. This has to be challenged.

Response: This is indeed a thoughtful point. Although HUVECs are classical models for in vitro investigations of endothelium and the permeability of both intestinal endothelium and HUVECs are prone to VEGF stimulation (Langer et al. JCI. 2019; Huang et al. Nat Microbiol. 2019), the tissue-specific effect of spike RBD may still exist as suggested by previous studies (Binion et al. Gastroenterology. 1997). Besides, it has been reported that ACE2 could regulate gut vascular barrier by bone marrow-derived cells via the PGN/MyD88/ARNO/ARF6 pathway (Duan et al. Circ Res. 2019). In addition to ACE-dependent modulation of the endothelial integrity, virus infection could also damage endothelium thus inducing vascular leakage. In agreement with your comments, we have discussed these possibilities in the Discussion section (P12, L8-12).

7. It is not either clear how effective is bevacizumab in mice.

Response: The effectiveness of bevacizumab in mice is indeed debatable. Therefore, we performed the intervention experiment using mouse VEGF B/N antibody (Bock et al. Invest Ophthalmol Vis Sci. 2009), which also effectively blocked spike RBD-mediated VEGF production, hyper-permeability and tissue inflammation (Fig EV5C-F).

Reviewer #2

1. The clinical data in Figure 1 is based on a relatively small number of patients. While some aspects of the data are certainly hard to expand (particularly the GI tract biopsies), the serologic analysis of VEGF levels and GI symptoms or disease progression could be easily expanded, or recapitulated in an independent and separate cohort of patients. I believe this to be critical to confirm the validity and reproducibility of this key aspect of the study. In addition, this would allow for other analyses, like longitudinal evaluation of VEGF during disease progression, and correlation of VEGF with other serologic markers of disease severity (CRP, ferritin, d-dimers).

Response: Thank you for understanding the difficulty to obtain patient biopsies. With our best effort, we have expanded an independent cohort for the serologic analysis of VEGF levels, and obtained similar results as the previous ones (Fig 1D and Fig EV1C-E). To be noted, these two batches analysis of VEGF were measured on two different technical platforms with Luminex AssayHuman XL Cytokine Discovery Premixed Kit (R&D) for the previous cohort and ELISA kit (Cat. No. ab222510, Abcam) for the supplementary cohort, and both yielded similar results. Moreover, the correlation analysis between VEGF and other serologic markers of disease severity, such as CRP, D-dimers and procalcitonin, were provided in the revised Table 2 and Table EV1.

2. The spike protein of SARS-CoV2 does not bind well to murine ACE2; from reading the materials and methods it is not clear how this was addressed experimentally in this paper. Did the studies involve transgenic hACE2 mice (does not seem to be the case from is written in the methods section)? Did the studies use mouse adapted spike RBD sequences (for example, see this paper <https://doi.org/10.1016/j.ebiom.2021.103381>)? Otherwise, it is hard to make sense of the story.

Response: Thank you for raising this important point. We have made several attempts to detect the affinity of spike RBD to murine ACE2 using different techniques and have also repeated the spike-mediated effects using hACE2 mice. The detailed results and evidence from literature are discussed below:

(1) The ability of spike RBD to bind to murine ACE2 is controversial. Based on the co-IP experiment, we and others detected an association between these two proteins (Unpublished Fig 2; Zhao et al. J Virol. 2020; Li et al. J Virol. 2004). In addition, other groups have obtained a possible binding of spike S1 or ECD to murine ACE2 by co-IP or SPR (Kuba et al. Nat Med. 2005; Gu et al. Front Immunol. 2021). Therefore, several published works have used wildtype mice for the study of spike-induced effects (Kuba et al. Nat Med. 2005; Raghavan et al. Front Cardiovasc Med. 2021; Shin et al. Int J Biol Sci. 2021; Wang et al. Acta Pharmacol Sin. 2021; Gu et al. Front Immunol. 2021).

- (2) However, some evidence using other techniques support against the binding of spike RBD to murine ACE2. Our data using Surface plasmon resonance (SPR) and biolayer interferometry (BLI) also failed to detect the direct binding of these two components (Unpublished Fig 3).
- (3) We consider that whether the effect of spike RBD is through or fully through ACE2 is beyond the scope of this work. The fact that mice with either murine ACE2 or human ACE2 made similar impact (Fig 2C-H and Fig EV3 A, B, D-F) might hint that an alternative receptor or, more likely, a co-receptor of ACE2 is involved (Nguyen et al. Nat Chem Biol. 2022; Gu et al. bioRxiv. 2020). However, we do believe that ACE2 plays a role in this regulation, since when we knocked down ACE2 alone the downstream pathway changed in a similar way as spike RBD stimulated (Fig EV4H); also, the spike-mediated ERK activation was abolished when ACE2 was depleted (Fig EV4I). Overall, the message conveyed by the current work concentrates on that the spike RBD can induce the production of VEGF in the enterocytes thus inducing permeability and inflammation in small intestine. Detailed mechanisms underlying this effect would worth further investigations (P12, L19-P13, L5).

3. Many of the changes in the ERK and other pathways that are shown by western blot are subtle; at times, the quantification shown in bar graphs is not representative of the western blots shown (see for example pMEK in Fig 3H). Moreover, the authors interpret Fig S3D and E as evidence that Spike RBD mediates its effects through ACE2 and not through Tmprss2, using siRNA experiments. However, the experiments seem to be simply an analysis of these kinases after siRNA of these receptors and do not include Spike RBD treatment, so they simply speak to the impact of those receptors on these pathways. Overall, these data need to be significantly strengthened - showing data points in the bar graphs would help, also it is noted that the pERK changes are most dramatic in duodenal tissue in vivo (Fig S4A) and this seems to be a more robust and physiologically significant model than cells in culture. The authors are encouraged to use this model for many of the data shown in 3C and 3H. In addition, as alluded to already, the siRNA experiments need Spike-RBD treatment (Figs S3D, E).

Response: Thank you for these valuable points. We have made pertinent investigations and the results are discussed as follows:

- (1) We corrected our representative WB graph and quantification for MEK, pMEK, P90RSK and p-P90RSK (Fig 3I).
- (2) To further prove that the Ras-Raf-MEK-ERK pathway is indeed activated by spike RBD, we included spike RBD treatment group in addition to siACE2. The results demonstrated that upon knockdown of ACE2, the phosphorylation of ERK induced by spike RBD did not further increase (Fig EV4I), indicating that ACE2

is essential to activate the spike RBD-mediated Ras-Raf-MEK-ERK signaling pathway in the enterocytes.

- (3) As for TMPRSS2, since knockdown of this co-receptor did not alter the EK pathway we did not include this in our current manuscript (Unpublished Fig 4).
- (4) For the purpose of confirmation, we examined the expression changes of proteins in the Ras-Raf-MEK-ERK pathway in hACE2/B6J mice stimulated by spike RBD (Unpublished Fig 5).

4. The experiments using Bevacizumab in vivo looking at gut permeability are quite nice, and one of the stronger parts of the paper. The story would be a lot stronger if the same experiment was performed in mice with COVID infection (either using a mouse adapted strain or ACE2 transgenic mice).

Response: We appreciate your positive comments. We thought the effect might be better evaluated by treating with mouse VEGF antibody, the data of which were included in Fig EV5C-F.

5. The following statement did not seem to correspond to the data that is being referenced: "Notably, interstitial edema was significantly correlated with disease type, acid reflux, total bilirubin, glutamic-pyruvic transaminase (ALT) and aspartate aminotransferase (AST) (Table 1 and Appendix Table S1), indicating vascular permeability change as a crucial factor in disease progression." Table 1 shows the correlation coefficients of various clinical parameters and VEGF levels and Appendix Table S1 shows the clinical characteristics of the cohort. Please clarify where the data actually is.

Response: Thank you for raising this point. We apologize for the mistake and have replaced Table 1 to reflect the correct data.

6. The Y axis of Figure 2F reads VEGF levels, but other cytokines are actually shown. Please correct.

Response: Thank you for pointing out this error. You probably meant Fig 2E. We have corrected the label for the Y axis in this figure and Fig EV2E.

7. ACE2 expression in HUVEC and Caco-2 (Fig 3A, B) needs to be shown in order to interpret this data.

Response: Excellent point. We detected a slightly decrease of ACE2 in spike-RBD treated Caco-2 cells but not in HUVECs (Fig EV4G), which was consistent with literature (Kuba et al. Nat Med. 2005) and the alternation of the downstream pathway. However, the mechanism behind the differential response of tissue/cell type-specific ACE2 to spike RBD is not clear.

References

Binion DG, West GA, Ina K, Ziats NP, Emancipator SN, Fiocchi C. Enhanced leukocyte binding by intestinal microvascular endothelial cells in inflammatory bowel disease. *Gastroenterology*. 1997 Jun;112(6):1895-907.

Bock F, Onderka J, Rummelt C, Dietrich T, Bachmann B, Kruse FE, Schlötzer-Schrehardt U, Cursiefen C. Safety profile of topical VEGF neutralization at the cornea. *Invest Ophthalmol Vis Sci*. 2009 May;50(5):2095-102.

Duan Y, Prasad R, Feng D, Beli E, Li Calzi S, Longhini ALF, Lamendella R, Floyd JL, Dupont M, Noothi SK, Sreejit G, Athmanathan B, Wright J, Jensen AR, Oudit GY, Markel TA, Nagareddy PR, Obukhov AG, Grant MB. Bone Marrow-Derived Cells Restore Functional Integrity of the Gut Epithelial and Vascular Barriers in a Model of Diabetes and ACE2 Deficiency. *Circ Res*. 2019 Nov 8;125(11):969-988.

Gu T, Zhao S, Jin G, Song M, Zhi Y, Zhao R, Ma F, Zheng Y, Wang K, Liu H, Xin M, Han W, Li X, Dong CD, Liu K, Dong Z. Cytokine Signature Induced by SARS-CoV-2 Spike Protein in a Mouse Model. *Front Immunol*. 2021 Jan 28;11:621441.

Gu Y, Cao J, Zhang X, Gao H, Wang Y, Wang J, Zhang J, Shen G, Jiang X, Yang J et al. Interaction network of SARS-CoV-2 with host receptome through spike protein. *bioRxiv*. 2020. <https://doi.org/10.1101/2020.09.09.287508> [PREPRINT]

Huang J, Kelly CP, Bakirtzi K, Villafuerte Gálvez JA, Lyras D, Mileto SJ, Larcombe S, Xu H, Yang X, Shields KS, Zhu W, Zhang Y, Goldsmith JD, Patel IJ, Hansen J, Huang M, Yla-Herttuala S, Moss AC, Paredes-Sabja D, Pothoulakis C, Shah YM, Wang J, Chen X. Clostridium difficile toxins induce VEGF-A and vascular permeability to promote disease pathogenesis. *Nat Microbiol*. 2019 Feb;4(2):269-279. doi: 10.1038/s41564-018-0300-x. Epub 2018 Dec 3. PMID: 30510170; PMCID: PMC6559218.

Kuba K, Imai Y, Rao S, Gao H, Guo F, Guan B, Huan Y, Yang P, Zhang Y, Deng W, Bao L, Zhang B, Liu G, Wang Z, Chappell M, Liu Y, Zheng D, Leibbrandt A, Wada T, Slutsky AS, Liu D, Qin C, Jiang C, Penninger JM. A crucial role of angiotensin converting enzyme 2 (ACE2) in SARS coronavirus-induced lung injury.

Langer V, Vivi E, Regensburger D, Winkler TH, Waldner MJ, Rath T, Schmid B, Skottke L, Lee S, Jeon NL, Wohlfahrt T, Kramer V, Tripal P, Schumann M, Kersting S, Handtrack C, Geppert CI, Suchowski K, Adams RH, Becker C, Ramming A, Naschberger E, Britzen-Laurent N, Stürzl M. IFN- γ drives inflammatory bowel disease pathogenesis through VE-cadherin-directed vascular barrier disruption. *J Clin Invest.* 2019 Nov 1;129(11):4691-4707.

Lal BK, Varma S, Pappas PJ, Hobson RW 2nd, Durán WN. VEGF increases permeability of the endothelial cell monolayer by activation of PKB/akt, endothelial nitric-oxide synthase, and MAP kinase pathways. *Microvasc Res.* 2001 Nov;62(3):252-62.

Li W, Greenough TC, Moore MJ, Vasilieva N, Somasundaran M, Sullivan JL, Farzan M, Choe H. Efficient replication of severe acute respiratory syndrome coronavirus in mouse cells is limited by murine angiotensin-converting enzyme 2. *J Virol.* 2004 Oct;78(20):11429-33.

Nguyen L, McCord KA, Bui DT, Bouwman KM, Kitova EN, Elaish M, Kumawat D, Daskhan GC, Tomris I, Han L, Chopra P, Yang TJ, Willows SD, Mason AL, Mahal LK, Lowary TL, West LJ, Hsu SD, Hobman T, Tompkins SM, Boons GJ, de Vries RP, Macauley MS, Klassen JS. Sialic acid-containing glycolipids mediate binding and viral entry of SARS-CoV-2. *Nat Chem Biol.* 2022 Jan;18(1):81-90.

Raghavan S, Kenchappa DB, Leo MD. SARS-CoV-2 Spike Protein Induces Degradation of Junctional Proteins That Maintain Endothelial Barrier Integrity. *Front Cardiovasc Med.* 2021 Jun 11;8:687783.

Shin HJ, Ku KB, Kim HS, Moon HW, Jeong GU, Hwang I, Yoon GY, Lee S, Lee S, Ahn DG, Kim KD, Kwon YC, Kim BT, Kim SJ, Kim C. Receptor-binding domain of SARS-CoV-2 spike protein efficiently inhibits SARS-CoV-2 infection and attachment to mouse lung. *Int J Biol Sci.* 2021 Aug 28;17(14):3786-3794.

Wang W, Li SS, Xu XF, Yang C, Niu XG, Yin SX, Pan XY, Xu W, Hu GD, Wang C, Liu SW. Danshensu alleviates pseudo-typed SARS-CoV-2 induced mouse acute lung inflammation. *Acta Pharmacol Sin.* 2021 Jul 15:1-10.

Wettschureck N, Strilic B, Offermanns S. Passing the Vascular Barrier: Endothelial Signaling Processes Controlling Extravasation. *Physiol Rev*. 2019 Jul 1;99(3):1467-1525.

Xu D, Fuster MM, Lawrence R, Esko JD. Heparan sulfate regulates VEGF165- and VEGF121-mediated vascular hyperpermeability. *J Biol Chem*. 2011 Jan 7;286(1):737-45.

Zhao X, Chen D, Szabla R, Zheng M, Li G, Du P, Zheng S, Li X, Song C, Li R, Guo JT, Junop M, Zeng H, Lin H. Broad and Differential Animal Angiotensin-Converting Enzyme 2 Receptor Usage by SARS-CoV-2. *J Virol*. 2020 Aug 31;94(18):e00940-20.

Unpublished Figure 1. The expression of phosphorylation of VEGFR2 (Try1175). The levels of phosphorylation of VEGFR2 (Try1175) in HUVECs co-cultured with the supernatant of Caco-2 cells that were treated with Control-Fc, Spike RBD-Fc, or ERK/VEGF inhibitors by Western blot.

Unpublished Figure 2. Co-IP characterization of the affinity between murine ACE2 and SARS-CoV2 spike RBD. C166 cell lysates were immunoprecipitated (IP) with Pierce Protein A/G Plus Agarose, and probed by Immunoblotting (IB) with the ACE2 and spike antibodies.

A

B

Unpublished Figure 3. SPR and BLI characterization of the affinity between murine ACE2 and SARS-CoV2 spike RBD.

A The murine ACE2s were immobilized on the CM5 chip, and sequentially tested the binding with serially diluted spike RBD.

B The murine ACE2s were immobilized on a sensor chip and sequentially tested the binding with spike RBD.

Unpublished Figure 4. TMPRSS2 regulation of the ERK pathway in enterocytes. The protein levels of TMPRSS2, Ras, c-Raf, pMEK, pERK and p-P90RSK in Caco-2 cells with TMPRSS2 knockdown by Western blot.

Unpublished Figure 5. SARS-CoV-2 spike RBD promotes the ERK pathway in hACE2-B6J mice. The protein levels of Ras, pERK, ERK and p-P90RSK in intestinal tissue of hACE2-B6J mice treated with Control-Fc and Spike RBD-Fc by Western blot.

1st Mar 2022

Dear Prof. He,

Thank you for the submission of your manuscript to EMBO Molecular Medicine. We have now received the enclosed reports from the two referees who re-reviewed your manuscript. As you will see, they are supportive of publication pending minor revisions, and I am therefore pleased to inform you that we will be able to accept your manuscript once the following points will be addressed:

1/ Referees' comments: please exert caution when interpreting Western Blot quantification, as suggested by Referee #1.

2/ Main manuscript text:

- Please remove the red text, and only keep in track changes mode any new modification.
- Please remove "data unpublished" (p. 6). As per our guidelines on "Unpublished Data", the journal does not permit citation of "Data not shown/data unpublished". All data referred to in the paper should be displayed in the main or Expanded View figures.
- Material and Methods:
 - o Clinical samples: please include a statement that informed consent was obtained from all subjects and that the experiments conformed to the principles set out in the WMA Declaration of Helsinki and the Department of Health and Human Services Belmont Report.
 - o Animal models: please indicate the gender of the mice.
 - o Ethics approval: Please place this section before Data Availability section.
- Data availability section: It is mandatory to deposit the raw sequencing data (unprocessed) in a public repository and the accession numbers and databases should be listed under 'Data Availability'.
- Author contributions: We note that you currently have together with you a total of 4 co-corresponding authors. Is that correct? Do you confirm equal contribution of these 4 people, able to take full responsibility for the paper and its content? Similarly, you currently list 4 co-first authors. While there is no limit per se to the number of first authors, 3 is rare, 4 even more so, and may not reflect as intended to the community.

3/ Figures:

Please provide in the figures or in their legends the exact p values, not a range.

4/ Thank you for providing source data. Please upload your .czi documents in a different format if possible.

5/ Checklist:

Please fill in the following sections:

- Human research participants
- Experimental study design and statistics -> sample size and inclusion-exclusion criteria
- Ethics: human participants -> authority granting ethics approval and informed consent
- Data availability: update as needed.

6/ Please note that all corresponding authors are required to supply an ORCID ID for their name upon submission of a revised manuscript. ORCID ID are currently missing for Prof. Ye Liu, Prof. Hong Shan.

7/ Thank you for providing The Paper Explained. I added minor modifications, please let me know if you agree or amend as you see fit. Please include The Paper Explained in the main manuscript file.

Problem:

COVID-19 patients with gastrointestinal (GI) symptoms tend to develop severe disease, but the underlying mechanism is unclear. VEGF is upregulated in the blood of COVID-19 patients, yet its association with the development of GI symptoms has not been explored.

Results:

VEGF level correlated with intestinal edema, GI symptoms and disease progression in COVID-19 patients. In an animal model mimicking intestinal inflammation upon stimulation with SARS-CoV-2 spike protein, we found that VEGF was over-produced in the duodenum prior to its ascent in the circulation, which led to hyperpermeability and systemic inflammation. Cell experiments demonstrated that the SARS-CoV-2 spike protein activated the Ras-Raf-MEK-ERK-VEGF pathway in enterocytes, which promoted VE-cad-mediated vascular permeability. Blocking either ERK or VEGF reversed hyperpermeability and alleviated intestinal inflammation stimulated by SARS-CoV-2 spike protein both in vitro and in vivo.

Impact:

This study identifies a possible route of SARS-CoV-2 spike-induced VEGF production in the GI tract, which leads to vascular permeability and inflammation. Mechanistically, it uncovers the spike-activated ERK/VEGF pathway in enterocytes. Translationally, this work provides potential targets for the treatment of GI symptoms, harnessing the disease progression of COVID-19.

8/ Thank you for providing a synopsis text. I included minor modifications, please amend as you see fit and upload your synopsis as an individual word file:

VEGF is a key factor in vascular permeability and inflammation. A correlation between VEGF and COVID-19-related GI symptoms was established: upon viral spike protein-induced ERK activation, VEGF was produced by the enterocytes, which led to inflamed and leaky gut.

- VEGF secretion positively correlated with the occurrence of GI symptoms and COVID-19 severity, making it a key factor to predict disease progression.
- VEGF production was induced by SARS-CoV-2 spike RBD through the Ras-Raf-MEK-ERK pathway in enterocytes.
- Blocking either ERK or VEGF relieved intestinal inflammation and leakage induced by spike both in vitro and in vivo.

Thank you for providing a nice synopsis picture. Please upload it as an individual png, jpeg, or tiff file of 550 px wide x 300-600 px high. Please make sure that the text remains legible.

9/ As part of the EMBO Publications transparent editorial process initiative (see our Editorial at <http://embomolmed.embopress.org/content/2/9/329>), EMBO Molecular Medicine will publish online a Review Process File (RPF) to accompany accepted manuscripts.

This file will be published in conjunction with your paper and will include the anonymous referee reports, your point-by-point response and all pertinent correspondence relating to the manuscript. Let us know whether you agree with the publication of the RPF and as here, if you want to remove or not any figures from it prior to publication.

I look forward to receiving your revised manuscript.

Yours sincerely,

Lise Roth

Lise Roth, PhD
Editor
EMBO Molecular Medicine

To submit your manuscript, please follow this link:

Link Not Available

***** Reviewer's comments *****

Referee #1 (Remarks for Author):

The revised version of the Zeng et al manuscript has been largely improved. The authors adequately replied to my comments by adding several new experimental data and controls.

minor comment: Although the phosphorylations of ERK, MEK and p90RSK are visible and quantified, because total levels are also varying, the authors should more cautious on such data.

Referee #2 (Comments on Novelty/Model System for Author):

One of my main concerns with the initial submission was that the study utilized SARS-CoV2 spike protein in conventional mice, who are not able to support active infection. Confirmatory experiments using transgenic mice expressing human ACE2 are now included which is an important addition to this reviewed version.

Referee #2 (Remarks for Author):

All my concerns have been addressed. I commend the authors for their thorough review and consideration of the comments made to the initial version of the manuscript.

The authors performed the requested editorial changes.

24th Mar 2022

Dear Prof. He,

Thank you for submitting your revised files. I am pleased to inform you that your manuscript is now accepted for publication and will be sent to our publisher to be included in the next available issue.

Thank you for depositing your data in a public repository. Please note that they should be publicly available before online publication (the indicated release date is currently 2024-03-12).

Please also confirm that you agree with the following synopsis:

VEGF is a key factor in vascular permeability and inflammation. A correlation between VEGF and COVID-19-related GI symptoms was established: upon viral spike protein-induced ERK activation, VEGF was produced by the enterocytes, which led to inflamed and leaky gut.

- VEGF secretion positively correlated with the occurrence of GI symptoms and COVID-19 severity, making it a key factor to predict disease progression.
- VEGF production was induced by SARS-CoV-2 spike RBD through the Ras-Raf-MEK-ERK pathway in enterocytes.
- Blocking either ERK or VEGF relieved intestinal inflammation and leakage induced by spike both in vitro and in vivo.

Additionally, to better promote your work among the Chinese readership, we would like to invite you to prepare a short summary of the manuscript in Chinese, which we will promote on the WeChat platform 'BioArt' with 240,000 followers.

If you are interested in taking up this opportunity, we recommend to cover the article very close to its online publication date. Thus, ideally we will very much appreciate it if you can send us a draft by replying to this invitation email within the next 7 working days. My colleague Jingyi (jingyi.hou@embo.org) will be happy to suggest any edits that may help to further improve the text. Please let us know within 3 days if you plan (or not) to contribute such a short summary in Chinese.

Below please find some general guidelines on how to prepare a summary. I have also included links to recent examples for your reference.

Thank you again and please let me know if you have any question. We look forward to reading your summary.

We also noted your suggestion to invite a News & Views or Commentary on your article.

Finally, we would like to remind you that as part of the EMBO Publications transparent editorial process initiative, EMBO Molecular Medicine will publish a Review Process File online to accompany accepted manuscripts. If you do NOT want the file to be published or would like to exclude figures, please immediately inform the editorial office via e-mail.

With kind regards,

Lise Roth

Lise Roth, PhD
Editor
EMBO Molecular Medicine